

# Numerical investigation of interaction between anticyclonic eddy and semidiurnal internal tide in the northeastern South China Sea

Liming Fan[1], Hui Sun[1*], Qingxuan Yang[1,2], Jianing Li[1]

[1]Frontier Science Center for Deep Ocean Multispheres and Earth System (FDOMES) and Physical Oceanography Laboratory/Sanya Oceanographic Institution, Ocean University of China, Qingdao/Sanya, 266100/572024, China
[2]Laoshan Laboratory, Qingdao, 266100, China

*Correspondence to*: Hui Sun (sunhui@ouc.edu.cn), Qingxuan Yang(yangqx@ouc.edu.cn)

**Abstract.** We investigate interaction between an anticyclonic eddy (AE) and semidiurnal internal tide (SIT) on the continental slope of the northeastern South China Sea (SCS), using a high spatiotemporal resolution numerical model. Two key findings are as follows. First, the AE promotes energy conversion from low-mode to higher-mode SIT; additionally, production terms indicate that the energy is also transferred from the SIT field to the eddy field at an average rate of -3.0 mW m$^{-2}$. Second, the AE can modify the spatial distribution of tidal-induced dissipation by both refracting and reflecting low-mode SIT. The phase and group velocities of the SIT are significantly influenced by the eddy field, resulting in a shift of the internal tidal rays to the north or south. These findings deepen our understanding of complex interactions between AE and SIT and of their impacts on energy conversion, wave propagation, and coastal processes.

## 1 Introduction

In the ocean, mesoscale dynamic processes (50-100 km) play a transitional role in energy cascade; among them, internal tide (IT) and mesoscale eddy (ME) are two of the most ubiquitous processes (Chelton et al., 2011; Müller et al., 2012; Faghmous et al., 2015; Zhao et al., 2016). The former is generated by the interaction of barotropic tide (BT) and topography in a stratified ocean, deriving about 1 TW energy [1 TW=1012 W] from BT globally (Egbert and Ray, 2003). The latter is often generated by hydrodynamic instabilities of background currents (Zhang et al., 2016), and accounts for about 80% of the total oceanic kinetic energy (Ferrari and Wunsch, 2009). Moreover, both IT and ME have essential effects on deep-ocean mixing, heat and mass transports, ecological productivity, and so on (Munk and Wunsch, 1998; Wunsch, 1999; Da Silva et al., 2002; McGillicuddy et al., 2007; Stastna and Lamb, 2008; Sharples et al., 2009; Lermusiaux et al., 2010; Duda et al., 2014; Zhang et al., 2014).

The propagation and dissipation processes of ITs and MEs have been research foci in recent years. When the IT obtains energy from the large-scale BT, its low-mode part undergoes long-distance propagation (exceeding 1000 km) to redistribute the energy (Zhao, 2017; Alford et al., 2019). During the propagation, the low-mode IT is susceptible to the states of





background fields, leading to incoherence and nonstationarity (Savage et al., 2020). The high-mode IT, on the contrary, dissipates mainly near the generation site due to its short wavelength and large vertical shear (Vic et al., 2019). Eventually, the dissipation of IT will affect the temporal variability of oceanic meridional overturning circulation (Munk and Wunsch, 1998). Unlike ITs, MEs have significant impacts on meridional transport of mass and on zonal transport of heat during their

propagations (Wunsch et al., 1999; Zhang et al., 2014). In addition, the dissipation of MEs is linked to sub-mesoscale processes (Zhang et al., 2016; Yang et al., 2019).

Due to the comparable horizontal scales of IT and ME, their interaction occurs easily and becomes a hotspot. When the interaction occurs, IT and ME exchange energy with each other. On the one hand, IT can transfer energy to ME, causing mesoscale kinetic energy to alter (Barkan et al., 2017, 2021). For example, in the Southern Ocean, the eddy field receives

$2.2\pm0.6$ mW m$^{-2}$ energy from the internal wave field (with frequencies ranging from f to N and including tidal frequencies) through vertical shear (Cusack et al., 2020). On the other hand, the energy transfer from ME to the internal wave field can induce viscous effects, which can be parameterized by the eddy viscosity coefficient to improve the accuracy of IT forecasts (Polzin, 2010). Furthermore, and most notably, MEs can modulate the generation, propagation, and inner-modal energy redistribution of ITs. Because the ME has a significantly longer time scale than the IT, it is common to assume the ME as the

background field and then focus on the modulation of the IT by the ME when studying their interaction.

Regarding how ME affects the generation of IT, studies have shown that the interaction between BT and baroclinic eddy field can generate internal waves (Krauss, 1999), which are most efficiently generated when their horizontal scales are comparable (Lelong and Kunze, 2013). ME mostly modulates the propagation of IT in terms of refraction and scattering. IT is refracted as it passes through the ME field (which corresponds to a non-uniform propagation medium), thus changing its

propagation direction (Huang et al., 2018). Meanwhile, IT can scatter from mode 1 to mode 2 and higher modes, accompanied by an inner-modal redistribution of IT energy (Dunphy and Lamb, 2014; Clément et al., 2016; Dunphy et al., 2017).

The refraction and scattering of IT can be detected in in-situ observations (Huang et al., 2018; Löb et al., 2020). However, there are some geographic scale restrictions when using observed data. So, numerical simulation of the whole process of IT

propagation is an essential way to study interaction between IT and ME. Using numerical models, researchers have directly simulated interaction processes (Dunphy and Lamb, 2014; Zaron and Egbert, 2014). To examine the energy changes during the interaction more accurately, Kelly and Lermusiaux (2016) proposed a refined internal wave energy equation to quantify the effect of background flow on internal wave generation and propagation. This equation has been used in the Middle Atlantic Bight and Palau Island waters (Pan et al., 2021), suggesting that it is an effective tool to study the interaction.

The SCS is a large marginal sea in the western Pacific Ocean, where IT and ME are ubiquitous; they are particularly energetic in the northern SCS (Niwa and Hibiya, 2004; Jan et al., 2008; Cheng et al., 2009; Chen et al., 2011; Guo et al., 2012; Li et al., 2012; Lin et al., 2015; Zhang et al., 2016; Cao et al., 2017). The northern SCS is therefore an excellent area for investigating the interaction between IT and ME. Such knowledge of the interaction between IT and background flow (e.g., ME) will conduce to a better understanding of the energy budget among different dynamic processes in the study area.





Such study can also provide a reference for improving IT prediction and developing more reliable coupled ocean-climate
models.

The remainder of this paper is organized as follows. In section 2, we provide an introduction to the dataset used and the
energy equation of IT. Section 3 is divided into two parts: the first part examines the interaction between AE and SIT from a
dynamic and energetic perspective, and the second explores the impact of AE on the kinematic characteristics of SIT.
Contributions of interaction terms to SIT energy are discussed in section 4, followed by conclusions in section 5.

## 2 Data and methods

### 2.1 Data

We use the MIT general circulation model (MITgcm) LLC4320 (Lat-Lon-Cap) outputs (Marshall et al., 1997). The model
has a horizontal resolution of 1/48° (about 2 km at the equator) and 90 vertical layers (with a vertical resolution of about 1 m
at the surface and 30 m down to 500 m). The fine grid allows for more accurate characterization of topographic changes,
which directly affect the generation of IT (Niwa and Hibiya, 2004; Kelly et al., 2021). Note that LLC4320 uses a global-
scale LLC grid, so it does not need open boundaries (Menemenlis et al., 2021). The model can effectively simulate free-
propagating internal waves such as ITs, while regional models cannot because of weaker internal wave amplitude in the
simulated area than observed (Liu and Gan, 2016; Mazloff et al., 2020). The outputs are from a forward simulation without
any artificial intervention such as data assimilation, so it is reliable for diagnosing the energy equation (Cummings and
Smedstad, 2013). In addition, LLC4320 has been widely used to analyse basin-scale circulations, internal waves, and
mesoscale and sub-mesoscale processes (Rocha et al., 2016; Lin et al., 2020; Su et al., 2020; Goldsworth et al., 2021; Liu et
al., 2023; Zhang et al., 2023). Despite these advantages, LLC4320 has some shortcomings. It does not consider near-bottom
drag and dissipation caused by BT, resulting in the simulated IT being slightly stronger than the observed (Yu et al., 2019;
Buijsman et al., 2020). This weakness does not undermine the conclusions of this research, because we focus on
investigating the mechanisms of interaction between AE and SIT, rather than comparing with field observations.

### 2.2 Methods

By introducing background flow and modal decomposition, the tidal-averaged and depth-integrated energy equation of
mode-n IT is given as follows (Kelly and Lermusiaux, 2016a):

$$\nabla \cdot \langle \boldsymbol{F}_n \rangle + \sum_{m=0}^{\infty} \langle A_{mn} \rangle = \sum_{m=0}^{\infty} \langle C_{mn} \rangle + \sum_{m=0}^{\infty} \langle P_{mn}^S + P_{mn}^B \rangle + \langle D_n \rangle \,, \tag{1}$$

where $\langle \cdot \rangle$ denotes the average over several tidal cycles (62 h is used in this paper), and $\nabla$ denotes horizontal divergence. $\boldsymbol{F}_n$,
$A_{mn}$, $C_{mn}$, $P_{mn}^S$, $P_{mn}^B$, and $D_n$ are the mode-$n$ IT energy flux, advection by the background flow, topographic conversion,
shear production, horizontal buoyancy production, and dissipation terms, respectively. Detailed introduction to calculation
for each term is given in Appendix A. Eq. (1) includes interaction terms between background flow and IT. One of the terms



is the advection term ($\langle A_{mn}\rangle$), which characterizes the influence of background flow on IT propagation, but does not involve any energy transfer between them. Another term is the production ($\langle P_{mn}^{S} + P_{mn}^{B}\rangle$), which measures the exchanged energy between background flow and IT. Noted that the advection, conversion ($\langle C_{mn}\rangle$), and production terms all involve cross-modal exchange terms between modes $m$ and $n$, indicating inner-modal energy scattering. Besides, the energy dissipation term ($\langle D_n\rangle$) includes nonlinear wave-wave interactions (e.g., parametric subharmonic instability process), self-advection, and

numerical errors. (Appendix B verifies the reasonableness of the dissipation term.) The equation separates different modes to better evaluate changes in each IT mode, and explores scattering between different modes.

Note that Eq. (1) needs to satisfy the assumption of small-amplitude linear internal waves (Kelly and Lermusiaux, 2016a). We find that the Froude number ($F_r = U_0/c$, where $U_0$ is BT tidal current, and $c$ is phase speed) of the first three modes of SITs in the study area is less than 1, which means that the first three modes of SITs are applicable to Eq. (1).

Before using Eq. (1) for energy analysis, we need to extract ME and IT from the LLC4320 data. In this paper, the background flow (including MEs) is obtained by time averaging the LLC4320 data over 62 h; and the semi-diurnal signal is extracted using a 4th-order Butterworth filter method with the bandpass band of 1.73-2.13 cpd. Then, the baroclinic velocity of SIT is obtained by subtracting the depth average from the filtering results. For the calculation of pressure perturbation, readers are referred to Wang et al. (2016). To reduce the computational load, we select grid resolution of 1/24° for the

calculation, which is fine enough to distinguish the first five modes (Buijsman et al., 2020).

## 3 Results

The study area of the northeastern SCS is shown in Fig. 1a. We selected a period for analysis, corresponding to 130-170 days (including three spring tide moments). During this period, an active AE appeared on the western side of the Luzon Strait (LS), which passed through the continental slope of the northeastern SCS, and eventually dissipated nearshore (region R2 in

Fig. 1a). Both Fig. 1b and Fig. 1c show three spring tide moments during the 130-170 days, although the zonal semidiurnal barotropic tidal current from LLC4320 is slightly greater than that from TPXO-v9 (Egbert and Erofeeva, 2002), which is consistent with the conclusion of Yu et al. (2019). The difference between the spring tide moments in Fig. 1b-c may be due to the bandpass filtering, so we use the TPXO-v9 result to determine the spring tide moments, which occurred on days 134, 150, and 162, respectively.



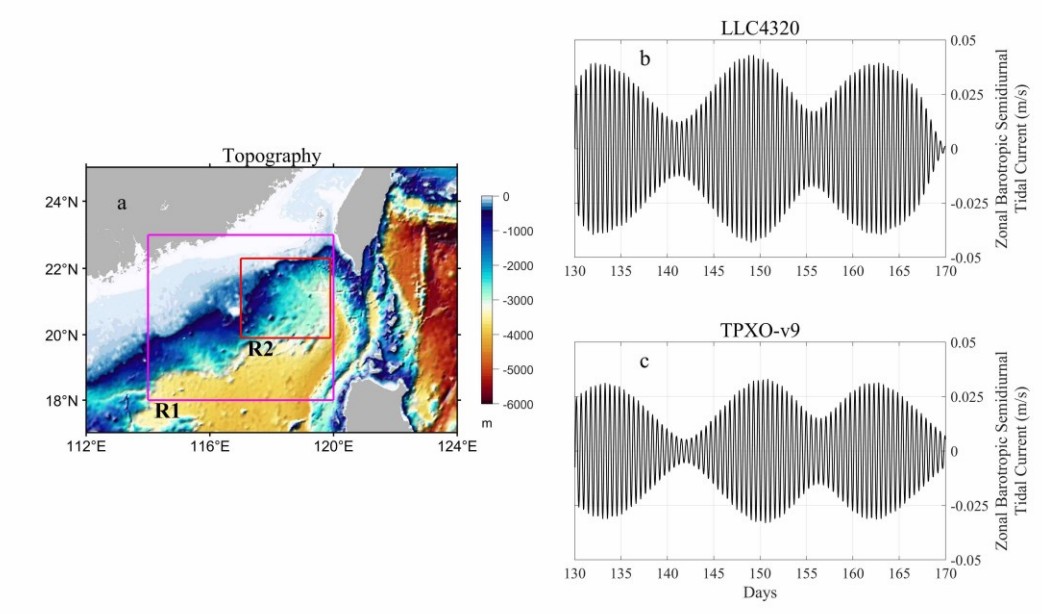


**Figure 1. (a)** Topographic distribution of the northern SCS, where R1 is the study area (114-120° E, 18-23° N) and R2 is the impact area of the AE (117.0-119.9° E, 19.9-22.3° N). **(b)** and **(c)** are the regionally averaged semidiurnal BT currents in region R2 extracted from the MITgcm LLC4320 and TPXO-v9 datasets, respectively.

### 3.1 Dynamics and energetics of the interaction between AE and SIT

To analyse the dynamics of the interaction between AE and the first three modes of SITs, we introduce the energy equation containing the background flow, as presented in Eq. (1). Eq. (1) reveals that $\nabla \cdot \langle \boldsymbol{F}_n \rangle$ can reflect the increase or decrease of IT energy, with positive value indicating a local source and negative value indicating a local sink. The increase or decrease of IT energy is mainly determined by three factors: topographic conversion term ($\langle C_{mn} \rangle$), background flow advection term ($\langle A_{mn} \rangle$), and energy exchange term between background flow and IT ($\langle P_{mn}^S + P_{mn}^B \rangle$).

In the following analysis, we primarily focus on the relative changes during the three spring tide moments. Since later analysis (Fig. 2d, Fig. 3d, and Fig. 4d) shows that changes in IT energy lag slightly behind changes in BT, we focus on days 137, 151, and 164.

### 3.1.1 Changes in the energy of SIT during the interaction

First, we analyse the impact of AE and SIT interaction on SIT energy. Figure 2 shows that the total energy (TE, the sum of 135   the depth-integrated horizontal kinetic energy (HKE) and available potential energy (APE)) of mode-1 SIT increases significantly during the AE period (days 146-162, which is determined through the change in area-integrated EKE; Fig. 2b), and decreases synchronously as the AE gradually dissipates on the continental slope (Fig. 2c). To quantify the energy input, we integrate the SIT energy flux on each side of region R2. The calculation shows that the energy input was 3.15 GW on day



137, increased to 4.31 GW on day 151, and then decreased to 3.27 GW on day 164, indicating an increase in SIT energy on
day 151.

Figures 2d-2f show that the energy of mode-1 SIT has three peaks (the peaks of SIT energy lag behind the spring tide moments by a few days because it takes time for the low-mode SITs to propagate from the generation source of the LS to R1 area) with a neap-spring tidal cycle. The neap-spring tidal cycle is influenced by the amplitudes of tidal constituents and the convergence or divergence of semidiurnal or diurnal tides, which are related to changes in the Moon's phase or declination
(Kvale, 2006). According to Fig. 2d, the TE of mode-1 SIT increased by 27% from day 137 to day 151, and decreased by 11% from day 151 to day 164. The changes in HKE (Fig. 2e) and APE (Fig.2f) are similar to that of TE, with an overall rising and then declining trend. In addition, the HKE-to-APE ratio ($r_E$) of progressive internal waves obeys

$$r_E = \frac{HKE}{APE} = \frac{\omega^2 + f^2}{\omega^2 - f^2},$$  (2)

where ω is the frequency of SIT and f is the local Coriolis frequency.

In Fig. 2f, the APE of mode-1 SIT in region R2 is consistent with the result of HKE/$r_E$, suggesting that the mode-1 SIT in the northeastern SCS satisfies the characteristic of free propagation (Zhao et al., 2010). We also compare the energy of mode-1 SIT with the results of Zhao and Qiu (2023), which verifies that the energy of mode-1 SIT in this region is larger in the north than in the south, and stronger in the east than in the west, suggesting that the LLC4320 data can simulate the generation and propagation of SIT well.

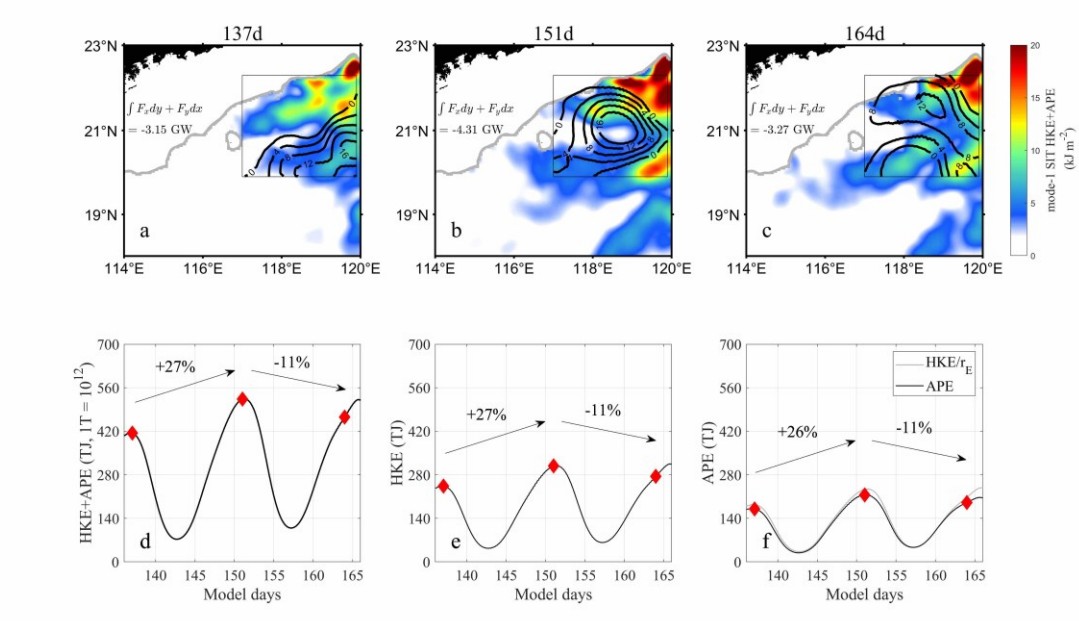


**Figure 2. (a-c)** Spatial distribution of mode-1 SIT energy on days 137, 151, and 164. Black contours represent sea-level anomalies (SLA; units: cm), and grey contours represent the depth of 250 m. **(d-f)** Time series of TE, HKE, and APE obtained from area integral over the region R2, with the red diamonds corresponding to days 137, 151, and 164, respectively. The grey curve in **(f)** is calculated using Eq. (2).



Figure 3 shows the energy of mode-2 SIT. The trend of increasing and then decreasing of TE is still evident, with a

significant feature that the TE is concentrated around the AE (Fig. 3b). According to area integral over the region R2, there

was 0.45 GW of energy input into the study area on day 137, which increased to 0.6 GW on day 151 before decreasing to

0.55 GW on day 164.

Based on $L_n \approx L_1/n^3$ (where $n$ is the mode number and $L_1$ is a characteristic propagation distance of mode-1 IT; Vic et

al., 2019) and $L_1 = 1000\text{-}1500$ km for SIT in the SCS (Xu et al., 2016), it can be inferred that the propagation distance of

mode-2 SIT is approximately 125-188 km. The results in Fig. 3a-c are consistent with the theory that mode-2 SIT can only

propagate westward to 117° E, because higher-mode ITs are more prone to dissipation and cannot travel long distances

(Nikurashin et al., 2011; Vic et al., 2019). Figure 3d indicates that the TE of mode-2 SIT first increased by 20% from day

to day 151, and then decreased by 10% from day 151 to day 164. Compared to changes in the TE of mode-1 SIT, the

enhancement ratio is smaller, but the reduction ratio remains roughly the same. HKE and APE have almost the same

temporal variation as TE (Fig. 3e and Fig. 3f). Note that in the selected area, the calculated $r_E$ is smaller than the theoretical

$r_E$, with the most significant difference occurring during spring tide. This implies that refraction or reflection likely occurred

for mode-2 SIT, leading to interference and deviation of calculated $r_E$ from its theoretical value (Martini et al., 2007;

Hamann et al., 2021).

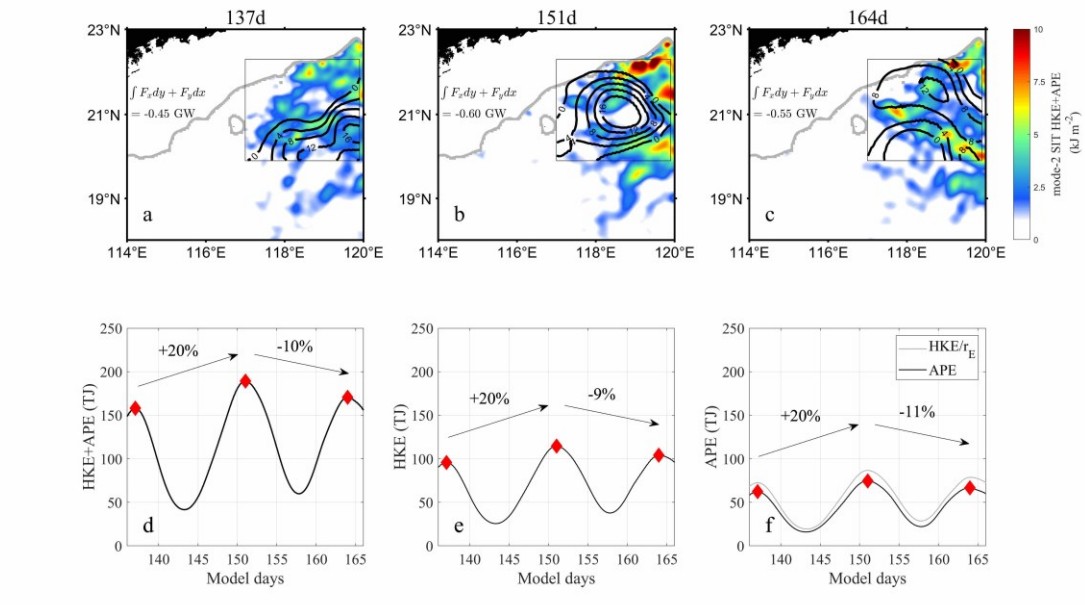

**Figure 3.** Same as Fig. 2, but for mode-2 SIT.

The energy of mode-3 SIT is shown in Fig. 4. The TE is not distributed in stripes or large patches as in the first two

modes, but in dispersed small blocks, implying that the propagation distance of mode 3 is shorter than that of the first two

modes. The theoretical propagation distance of mode-3 SIT is about 37-56 km according to the formula $L_3 \approx L_1/27$,





equivalent to the small blocks' horizontal scale. From the integral of energy flux in region R2, there was 0.05 GW energy

input into the study area on day 137, 0.01 GW on day 151, and 0.07 GW on day 164, with a trend of decreasing and then

increasing, which is opposite to the first two modes. Figure 4d shows that the TE of mode-3 SIT has an increasing-then-

decreasing trend (increased by 8% from day 137 to day 151 and decreased by 1% from day 151 to day 164), which is closer

to the change in HKE (Fig. 4e) rather than to the change in APE (Fig. 4f). Differences in temporal trends of HKE and APE

lead to significant variation between calculated $r_E$ and theoretical $r_E$. In summary, there existed local generation sources for

mode-3 SIT in region R2, since the TE of mode-3 SIT increased when the external energy input decreased on day 151

(though only by 0.01 GW).

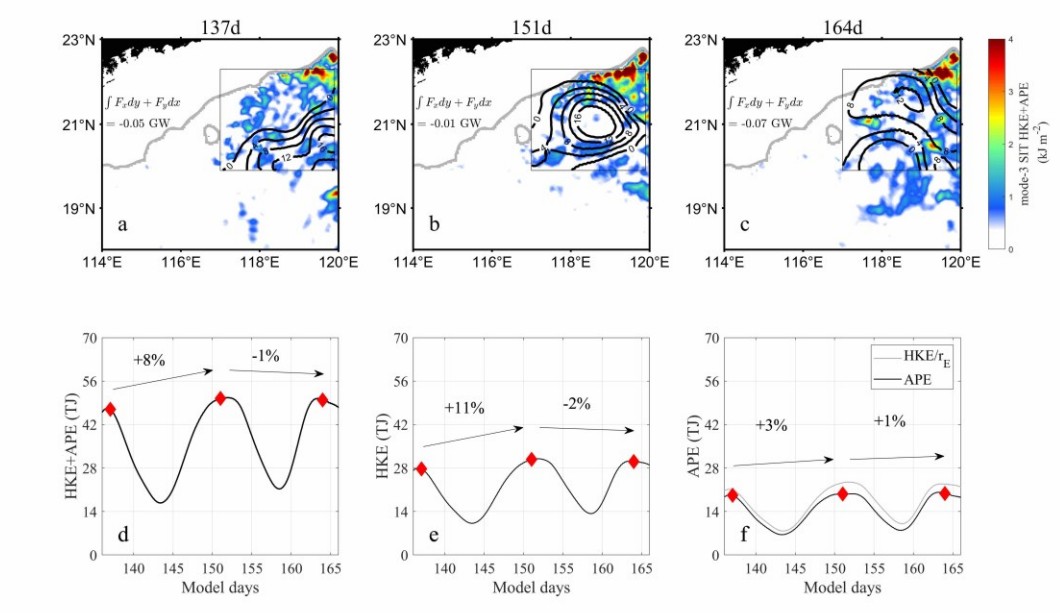

**Figure 4.** Same as Fig. 2, but for mode-3 SIT.

**Table 1.** Sectional integrated energy flux (EF) and total energy (TE) of the first three modes of SITs on days 137 and 151. The percentage
represents the relative change from day 137 to day 151.

|  | EF (GW) | | relative | TE (TJ) | | relative |
|---|---|---|---|---|---|---|
|  | 137d | 151d | change | 137d | 151d | change |
| Mode1 | 3.15 | 4.31 | 37% | 414 | 523 | 27% |
| Mode2 | 0.45 | 0.60 | 33% | 158 | 189 | 20% |
| Mode3 | 0.05 | 0.01 | -80% | 47 | 50 | 8% |

The impacts of the AE on SIT energy are outlined in Table 1. Day 137 serves as a reference when there was no effect from

the AE, and day 151 is taken as the result under the influence of the AE. For the external input part (EF), mode 1 grew by




37% during the AE period, mode 2 increased by 33%, while mode 3 decreased by 80%. The external energy input of the SIT came mainly from the LS (the generating site); in comparison, the contribution of the AE was negligible. Regarding TE, mode 1 increased by 27%, mode 2 increased by 20%, and mode 3 had an increase of 8%. If the dissipation of the SIT (e.g., topographic drag friction and wave-wave interaction) is assumed to be a fixed proportion of the EF, the relative change for the TE of mode-1 SIT should be 37%; however, it was only 27%. This suggests that another type of interaction may have inhibited the overall growth of mode-1 TE. For example, transferring energy to higher-mode SIT ultimately led to a decrease in low-mode TE (Dunphy and Lamb, 2014; Dunphy et al., 2017; Huang et al., 2018). Convincing evidence is that mode-3 SIT realized an inverse increase in TE when the external energy input was reduced.

### 3.1.2 Contribution of topographic conversion

First, we analyse the contribution of the topographic conversion term. The spatial distributions of the first three modes on day 151 are shown in Fig. 5a-c. The topographic conversion term for mode 1 is mostly negative, with most of the local minimums occurring at water depths of 250-2000 m. The situation is different for mode 2; it is primarily positive in the areas where mode 1 has local minimums, and it also contains regions with negative values. The distribution of mode 3 is similar to that of mode 2, except that its positive values cover a larger region than mode 2's, and its negative values have smaller amplitude and coverage than mode 2. Figures 5a-5c indicate that during the AE period, topographic conversion manifested as low-mode SIT scattering toward higher modes.

Figures 5d-5f show area integral over the region R2. Figure 5d shows the results of mode 1, with a trough during the AE period on day 151. Calculations indicate that a total of 0.93 GW energy was scattered into higher-mode SIT on day 151, while only 0.58 GW was scattered on day 137. Thus, the AE enhanced the topographic scattering of mode 1 by 60%. This might be one of the energy sinks of mode-1 SIT, resulting in a smaller relative increase in its TE than in its EF during the AE period, as shown in Fig. 2 and Table 1. Figure 5e shows the results for mode 2, where the integrated energy was comparable on day 151 (0.25 GW) and day 137 (0.21 GW), indicating that mode 2 gained energy from mode 1 while simultaneously scattering energy to higher modes, resulting in only a slight increase in its energy. Finally, Fig. 5f suggests that mode-3 energy increased significantly during the AE period (from 0.26 GW on day 137 to 0.44 GW on day 151), totalling an increase of 0.18 GW or a growth rate of 69%. This result is consistent with those in Fig. 4 and Table 1, which concludes that when the AE appeared within the region R2, there was a local energy source of mode 3, i.e., energy from low-mode topographic scattering.



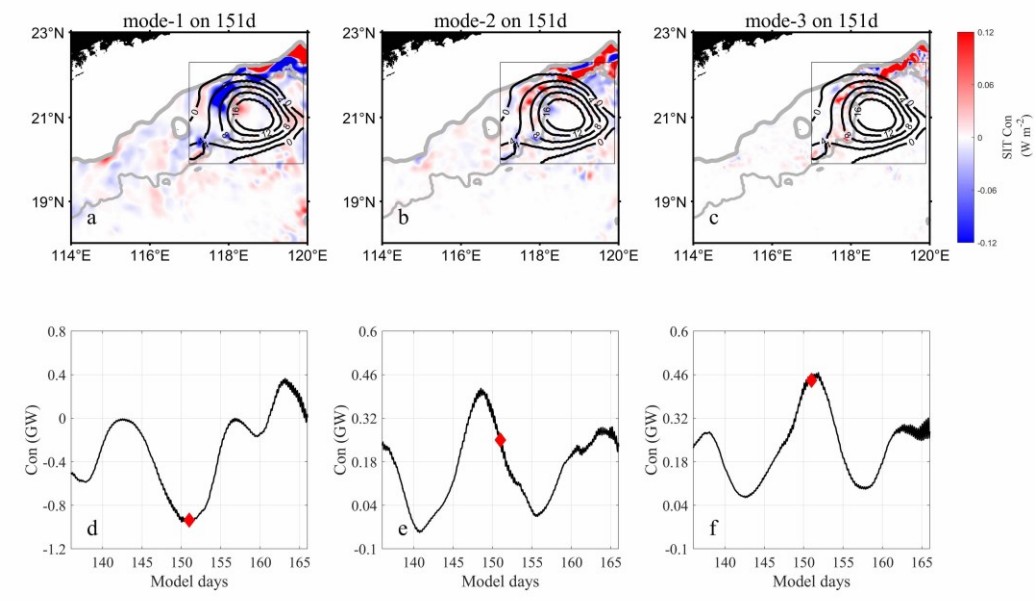

**Figure 5.** Spatial distribution of topographic conversion term for mode-1 to mode-3 SIT on day 151 in panels **(a-c)**. Panels **(d-f)** show the time series of their area integrals over the region R2. The red diamond indicates the result on day 151. The thick grey contour in **(a-c)** indicates the depth of 250 m, and the thin grey contour, the depth of 2000 m.

Figure 6 presents that $C_{mn}$ is antisymmetric ($C_{mn} = -C_{nm}, C_{mm} = 0$). Generally, the values of $C_{mn}$ for $m$=1:4 and $m<n$ are positive (except for $C_{1,5}$ on day 164), which means that low modes transfer energy to higher modes. At the same time, the energy conversion between adjacent modes was usually more significant than those between nonadjacent modes. Besides, the energy conversion between adjacent modes (such as $C_{1,2}$, $C_{2,3}$, $C_{3,4}$, and $C_{4,5}$) were all the largest on day 151 compared to days 137 and 164. This indicates that the AE promoted downscale energy transfer between different SIT modes and

efficiently scattered low-mode energy into higher modes (Hu et al., 2020; Löb et al., 2020), which is conducive to IT's turbulence dissipation process (Fernández-Castro et al., 2020).





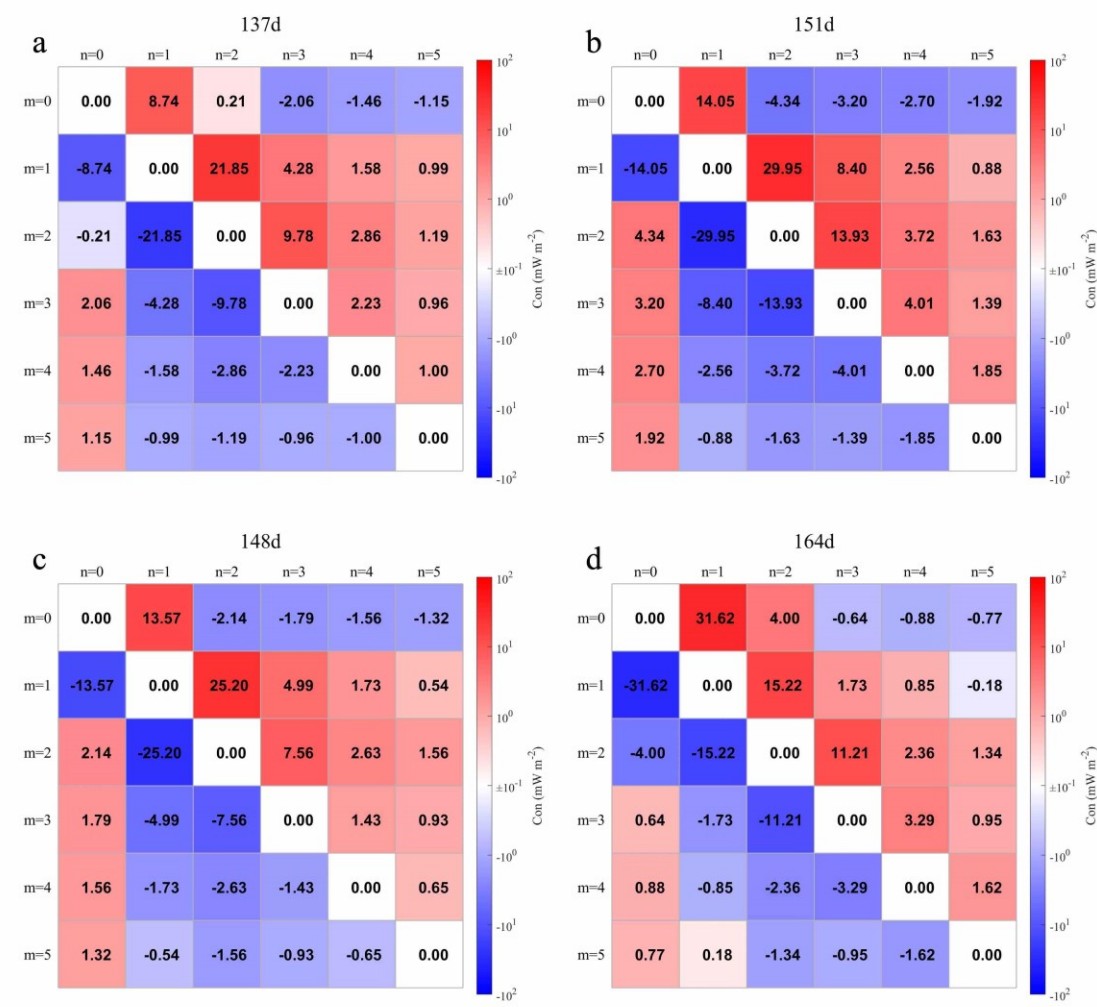

**Figure 6.** Distributions of topographic conversion term $C_{mn}$ (spatially averaged over the region R2 with water depths between 250 and 2000 m) in the domain of mode number on days 137, 151, 148, and 164. Subscripts $m$ and $n$ represent the work done by mode $m$ on mode $n$, respectively.

Figure 5e shows that the topographic conversion of mode 2 reached its maximum value on day 148 rather than on day 151. This is a result of the competition between two processes: acquiring energy from lower modes ($C_{0,2}$ and $C_{1,2}$) and scattering energy to higher modes ($C_{2,3}$, $C_{2,4}$, $C_{2,5}$). We calculate $\sum_{m=0}^{5}\langle C_{m,2}\rangle$ on day 148 based on Fig. 6c and obtain a value of 11.31 mW m$^{-2}$, which is larger than that on day 151 ($\sum_{m=0}^{5}\langle C_{m,2}\rangle$ = 6.33 mW m$^{-2}$ based on Fig. 6b). Note that on day 151 $C_{0,2-5}$ was negative, in contrast to the results obtained on seamounts by Lahaye et al. (2020), because most topographic critical parameters within the region R2 are less than 1 (Yang et al., 2016). Only when critical or supercritical topography is present, such as in the LS ($C_{0,1-5}$ in the LS are positive values, not shown), high-mode ITs will be generated in large

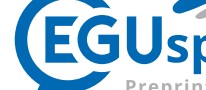

quantities (Wu et al., 2013; Xu et al., 2016). Therefore, high-mode SITs within the study area can only obtain energy from low-mode SITs.

### 3.1.3 Contribution of advection by the AE

Next, we analyse the impact of the advection term on SIT's energy changes. Using Eq. (A2), we find that the advection term consists of two parts, one part related to the baroclinic velocity of SIT, called $A_{mn}^u$, and the other related to the pressure perturbation of SIT, called $A_{mn}^p$. The results of $A_{mn}^u$ are shown in Fig. 7. As the mode number decreases, the amplitude of $A_{mn}^u$ increases, because the low-mode ITs have larger velocities and more energy (Liu et al., 2019). Figures 7a-7b present that the advection was most intense around the AE, with roughly alternating positive and negative distributions along the sea-level anomalies (SLA) contours, which is consistent with the findings of Dunphy and Lamb (2014). The domain integrals (Fig. 7d-f) show that positive and negative values vary symmetrically along the zero line, with peak or trough appearing during the AE period. Adding up the positive and negative values, we find that the advection term weakened the energy of mode 1 (note that the advection term appears on the left side of Eq. (1), so a positive value represents a decrease in IT energy ), and enhanced the energy of modes 2 and 3. The results for $A_{mn}^p$ (Fig. 8a-c) have similar spatial distribution patterns to those of $A_{mn}^u$, but with smaller amplitude, implying that the velocity component of the advection was dominant. This conclusion is also applicable when integrating over space (Fig. 8e-f). The effects of $A_{mn}^p$ on the first three modes of SITs are similar to those of $A_{mn}^u$.

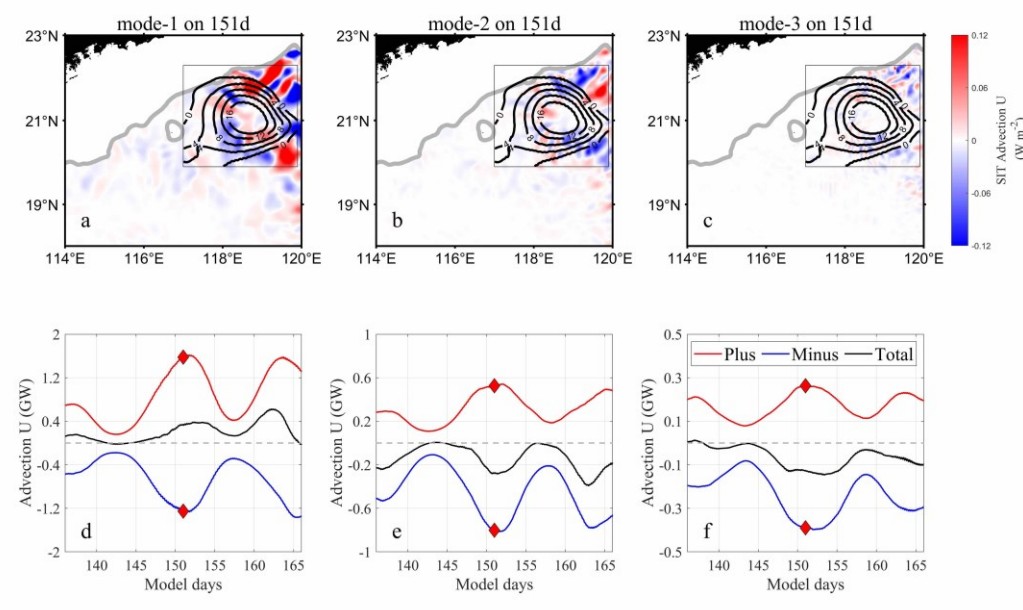

**Figure 7.** Same as Fig. 5, but for the advection term of the velocity component of SIT. The red, blue, and black curves in **(d-f)** represent the spatial integral results of positive, negative, and all values within the rectangle, respectively.



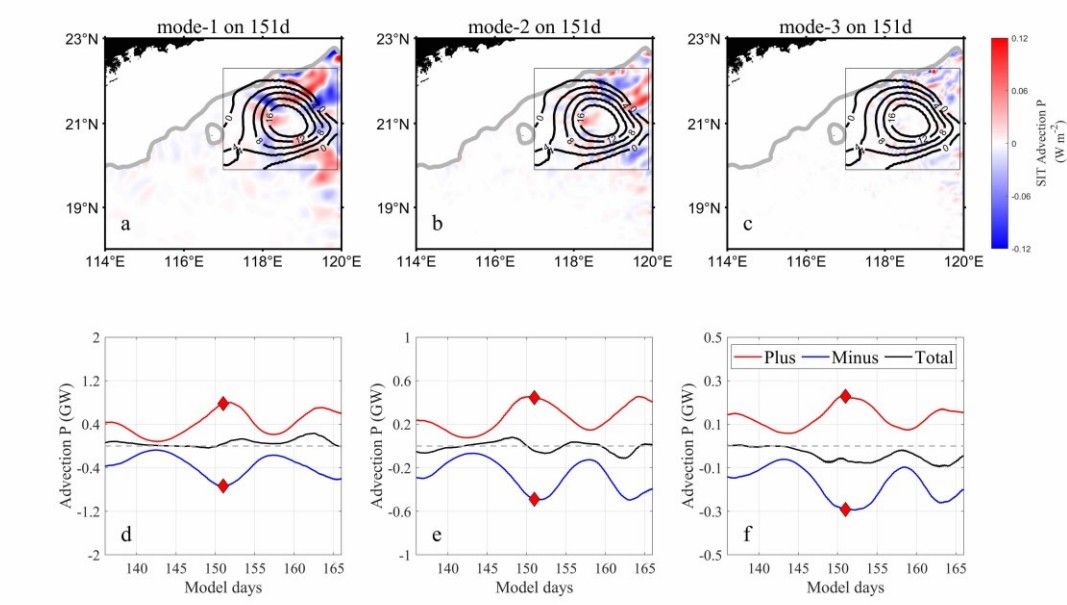

**Figure 8.** Same as Fig. 7, but for the advection term of the pressure component of SIT.

### 3.1.4 Contribution of energy exchange between AE and SIT

Finally, we analyse the impact of the energy exchange term on SIT's energy changes. The energy exchange term can be subdivided into horizontal shear production, vertical shear production, and horizontal buoyancy production (Fig. 9). The greater the mode number, from left to right in Fig. 9, the weaker the energy exchange. The magnitude of the vertical shear production is the largest of the three components, implying that the vertical shear of the eddy field dominates the energy exchange between the AE and SIT. In terms of the R2 area integrals (Fig. 10), the temporal trend of mode 1 is similar to that

of the advection term (Fig. 7 and Fig. 8), with extremes during spring tides; mode 2 exhibits peak or trough only on day 151, and mode 3 has no prominent peak. During the AE period, shear production led to negative/upscale energy exchange (from the IT field to the eddy field); conversely, buoyancy production mainly induced positive/downscale energy exchange (from the eddy field to the IT field). Overall, there was a net energy exchange from the IT field to the eddy field during the AE period, with a regional average rate of -3.0 mW m$^{-2}$ on day 151 (the sum of the energy exchange terms of the first three

modes divided by the square of R2, $\sum_{n=1}^{3}(P_n^s + P_n^b)/S$). This is quite close to the result of -2.2±0.6 mW m$^{-2}$ obtained by Cusack et al. (2020) in the Southern Ocean, which suggests that the IT field may act as an energy source for the eddy field.



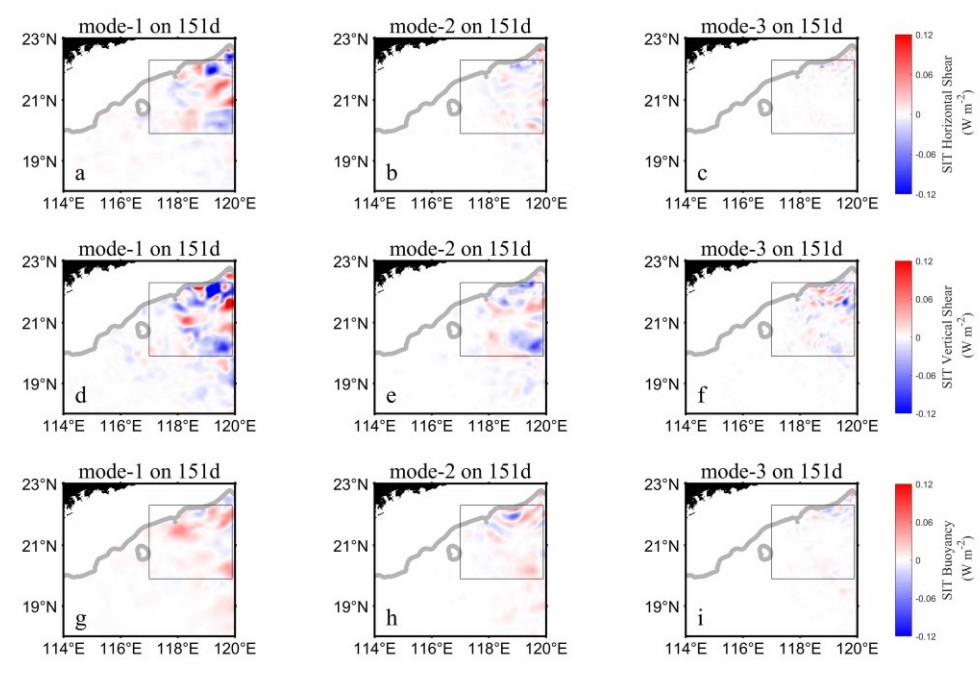

**Figure 9.** Spatial distributions of the first three modes of SITs on day 151, with **(a-c)** representing horizontal shear production, **(d-f)** representing vertical shear production, and **(g-i)** representing horizontal buoyancy production.

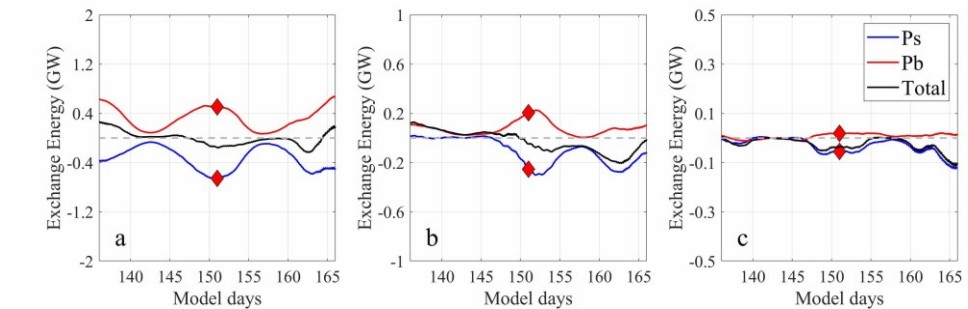


**Figure 10.** Time series of spatial integration for the first three modes **(a-c)**, with Ps representing the sum of the horizontal and vertical shear production terms, Pb representing the horizontal buoyancy production, and Total being their sum.

### 3.2 Effects of the AE on kinematic properties of SIT

#### 3.2.1 Impact of the AE on low-mode SIT reflection over continental slope

During the propagation of low-mode SITs, different types of continental slope topography may be encountered. According to

the topographic steepness parameter,





$$\gamma = \frac{|\nabla H|}{\sqrt{\frac{\omega^2 - f^2}{N^2 - \omega^2}}}, \tag{3}$$

where H is water depth, topography can be classified into three types: subcritical topography ($\gamma<1$), where IT can continue to propagate and shoal onto the continental slope; critical topography ($\gamma=1$), where IT is captured by the topography and

subsequently breaks; and supercritical topography ($\gamma>1$), where internal tidal rays are reflected by the continental slope (Pedlosky, 2003; Kelly et al., 2013; Legg, 2014).

Here, we focus on the energy reflection of mode-2 SIT on a supercritical continental slope, as Fig. 3b shows that part of its energy is reflected on the slope. First, we adopt a directional Fourier filter method (DFF; Gong et al., 2021) to decompose the baroclinic velocity ($\boldsymbol{u}$) and pressure perturbation ($p$) of mode-2 SIT into incident and reflected components:

$$\begin{cases} \boldsymbol{u} = \boldsymbol{u}_i + \boldsymbol{u}_r \\ p = p_i + p_r \end{cases}, \tag{4}$$

where subscripts $i$ and $r$ represent incident and reflected components, respectively. Then, the energy fluxes of incident and reflected waves are calculated as follows:

$$\begin{cases} \boldsymbol{F}_i = \boldsymbol{u}_i p_i \\ \boldsymbol{F}_r = \boldsymbol{u}_r p_r \end{cases}, \tag{5}$$

Wang et al. (2020) concluded that the DFF method is suitable and most effective for processing 3D numerical model outputs

after comparing three different decomposition methods.

Figure 11a depicts the total energy flux of mode-2 SIT on day 151 (under the influence of the AE), demonstrating that the energy flux propagating northwestward may be reflected on the continental slope (near 119° E, 22° N). To verify this reflection phenomenon, we examine the energy fluxes along the incident and reflected directions in Fig. 11b-c, respectively. According to the incident energy flux integrated along section S1, a total of 0.70 GW mode-2 SIT propagated toward the

continental slope (Fig. 11b). The reflected energy flux integrated along section S2 implies that there was a total of 0.31 GW mode-2 SIT energy being reflected off the continental slope with a reflection coefficient of 44% (Fig. 11c). However, there were 0.31 GW along section S1 and 0.10 GW along section S2, with a reflection coefficient of just 32% on day 137 (without the AE). Previous studies showed that the magnitude of the reflection coefficient is closely related to the steepness of the continental slope; for example, on the supercritical Tasmanian continental slope, the reflection coefficient of SIT can exceed

60%, forming a distinct standing wave structure (Johnson et al., 2015; Klymak et al., 2016; Zhao et al., 2018). The spatial distribution of topographic steepness parameters during the AE period (Fig. 11d) also indicates that the continental slope was mainly supercritical for SIT, which favored the reflection of SITs.



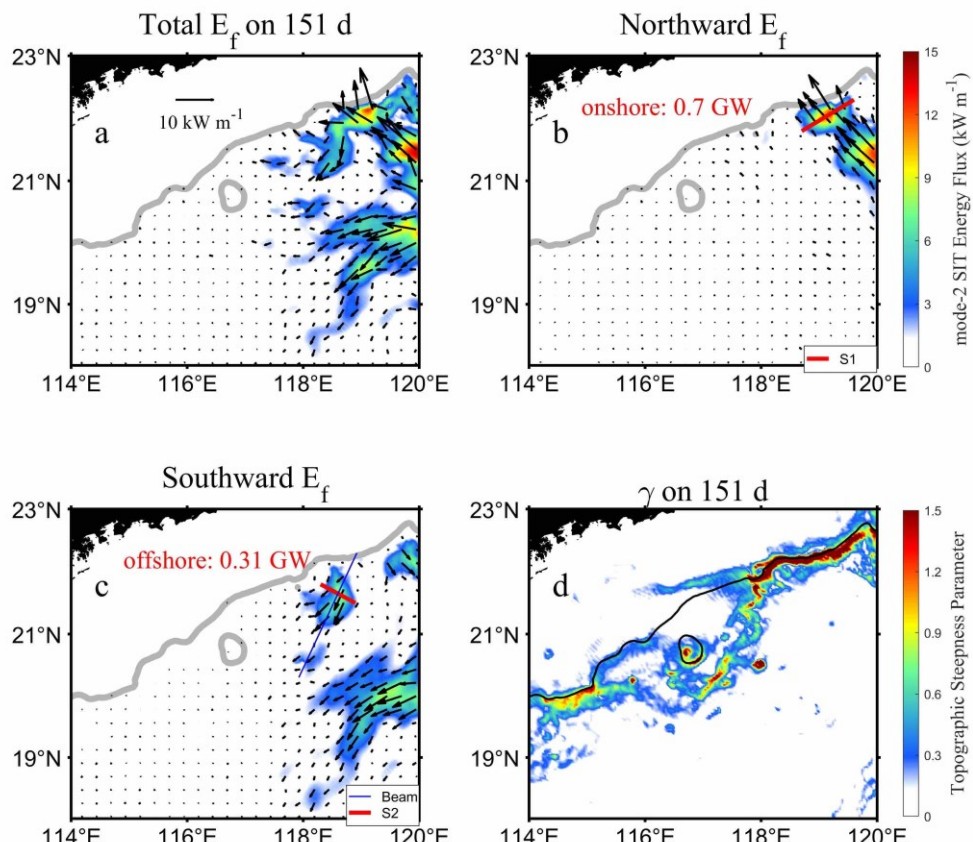

**Figure 11.** The energy flux of mode-2 SIT on day 151 in **(a)**, **(b)**, and **(c)** for total, northward, and southward, respectively. The energy
fluxes integrated along sections S1 and S2 are labelled as onshore and offshore values, respectively. The topographic steepness parameter
for SIT on day 151 is presented in **(d)**.

We find mode-2 SIT radiated from the shallow supercritical continental slope to the inner ocean using the theoretical ray
path. To analyse how the AE promoted the reflection of SIT, we calculate the theoretical wave rays along the blue line in
Fig. 11c. Figure 12 shows that isopycnals above 1000 m deepened on day 151 compared to that on day 137; the isopycnal of
1026 kg m$^{-3}$ deepened most. The topographic steepness parameter obtained from Eq. (3) also shows differences between the
two periods: the nearshore slope (0-20 km) had a larger steepness parameter on day 151 than on day 137.

Through theoretical wave ray calculations, we find that mode-2 SIT was reflected from the shallow supercritical
continental slope to the deep-sea basin of the SCS. Although the paths of the wave rays were similar, the propagation
distance of wave rays on day 151 was somewhat longer than that on day 137. Besides, the mode-2 baroclinic velocities
visually demonstrate that the energy of the reflected mode-2 SIT was significantly enhanced under the impact of the AE
(maximum amplitude of 0.16 m s$^{-1}$ on day 151 and 0.09 m s$^{-1}$ on day 137), indicating that the AE promoted SIT reflection.





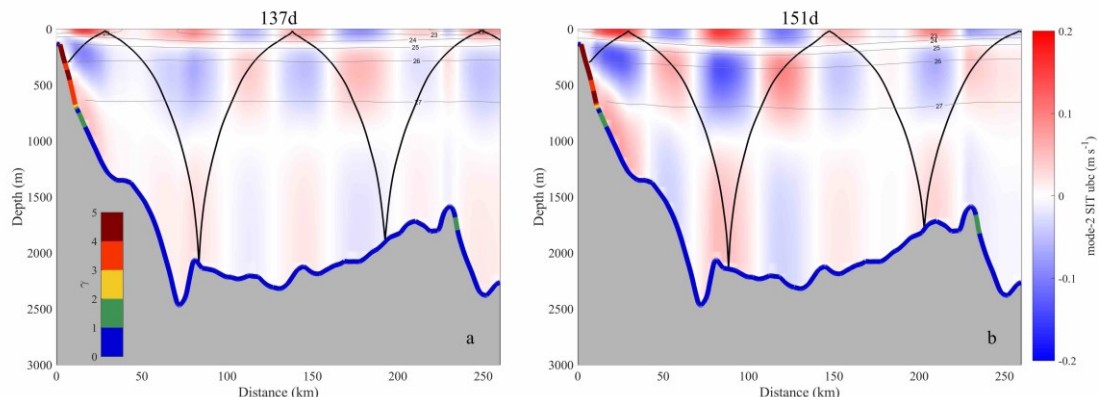

**Figure 12.** The baroclinic velocity of mode-2 SIT on days **(a)** 137 and **(b)** 151 along the Beam section shown in Fig. 11c. Black curves are theoretical wave rays, and grey contours are isopycnals (ρ-1000). The colors marked along the seafloor represent the values of the topographic steepness parameter.

### 3.2.2 Effect of the AE on low-mode SIT propagation and refraction

Here, we study the impact of the AE on SIT propagation and refraction during the AE period. As is well known, the greater the mode number of IT, the shorter the propagation distance. For high-mode ITs (mode number ≥4), their propagation distances cannot exceed 1/2° (about 50 km), so they contribute to near-filed dissipation at their generation sites (St. Laurent and Garrett, 2002; Vic et al., 2019). Unlike high-mode ITs, low-mode ITs dissipate in far-filed regions and are therefore more susceptible to interference from background oceanic processes during their propagation. As a result, it is challenging to predict the propagation of low-mode SITs accompanied by the occurrence of tidal nonstationarity and incoherent processes (Whalen et al., 2018; Savage et al., 2020). The spatial distribution of tidal-induced mixing in the ocean is affected by the propagation path and distance of SIT, which is critical to the formation of deep-sea overturning circulations (You et al., 2022, 2023; Zhou et al., 2023).

The phase speed is a useful variable for studying the propagation of SIT. When considering the influence of background flow, we usually obtain the eigenvalues by solving the Taylor-Goldstein equation (Smyth et al., 2011). However, the equation does not take into account the Coriolis effect; hence, a correction for Earth rotation effects is introduced (Zhao et al., 2010), which yields Eq. (6),

$$
\begin{cases}
\dfrac{d^2\Phi_n}{dz^2} + \left[\dfrac{N^2}{(U-c_n^U)^2} - \dfrac{d^2U/dz^2}{U-c_n^U}\right]\Phi_n = 0 \\
c_p^U = \dfrac{\omega}{\sqrt{\omega^2-f^2}}c_n^U
\end{cases}, \tag{6}
$$

where $\Phi_n$ indicates the eigenfunction of vertical structure, subscript $n$ represents the $n$th mode, and $N^2$ indicates the buoyancy frequency squared. $c^U$ and $c_p^U$ are the eigen speed and phase speed, respectively, considering both the stratification and background flow. If we exclude the impact of the background flow, Eq. (6) can be simplified to Eq. (7),



$$\begin{cases} \frac{d^2\Phi_n}{dz^2} + \frac{N^2}{c_n^2}\Phi_n = 0 \\ c_p = \frac{\omega}{\sqrt{\omega^2 - f^2}}c_n \end{cases},$$

(7)

where $c$ and $c_p$ are the eigen speed and phase speed, respectively, considering only the stratification.

We studied the influence of the AE on the propagations of mode-1 and mode-2 SIT. Figure 13 shows the SIT phase speed calculated by using Eq. (7), considering only the AE-associated stratification. Figures 13a-13c show that when the AE appeared, it altered the buoyancy frequency in the ocean, causing changes in SIT phase speed. The AE enhanced the mode-1 SIT phase speed, with a maximum increase of 0.35 m s$^{-1}$ and an average increase of 0.09 m s$^{-1}$ within the region R2 (Fig.

13c). The effect of the AE on mode 2 is similar to that on mode 1, but the overall increase was smaller, with maximum and average increases of 0.12 and 0.02 m s$^{-1}$, respectively (Fig. 13f).

To more intuitively describe how SITs propagate under different background conditions, we use Park and Farmer's (2013) method to calculate the ray paths for SITs. At the same time, to consider the interactions among mode 1, mode 2, and the AE, we select two ray starting points. The north starting point is located at (119.8° E, 21.5° N), with an initial angle of 170°

(east corresponds to 0°, counterclockwise rotation). The south starting point is at (119.8° E, 20.5° N), and the initial angle is 185°.

During day 137, mode 1 propagated 193 km northwest from the north starting point before encountering topography (stop calculating when the water depth is less than 250 m); from the south starting point, it travels southwestward for 248 km (Fig. 13a). At the same time, mode 2 propagated northwestward for 135 km and southwestward for 138 km (Fig. 13d). During the

period of the AE (on day 151; Fig. 13b and Fig. 13e), the SIT propagated almost the same distance; however, the north ray was shifted by 5° clockwise and the south ray was shifted by 8° counterclockwise for mode 1, while the two rays of mode 2 showed little variation. These results suggest that although the AE can affect background stratification, its short influence distance (limited by AE diameter) did not significantly affect the propagation of the first two modes of SIT.





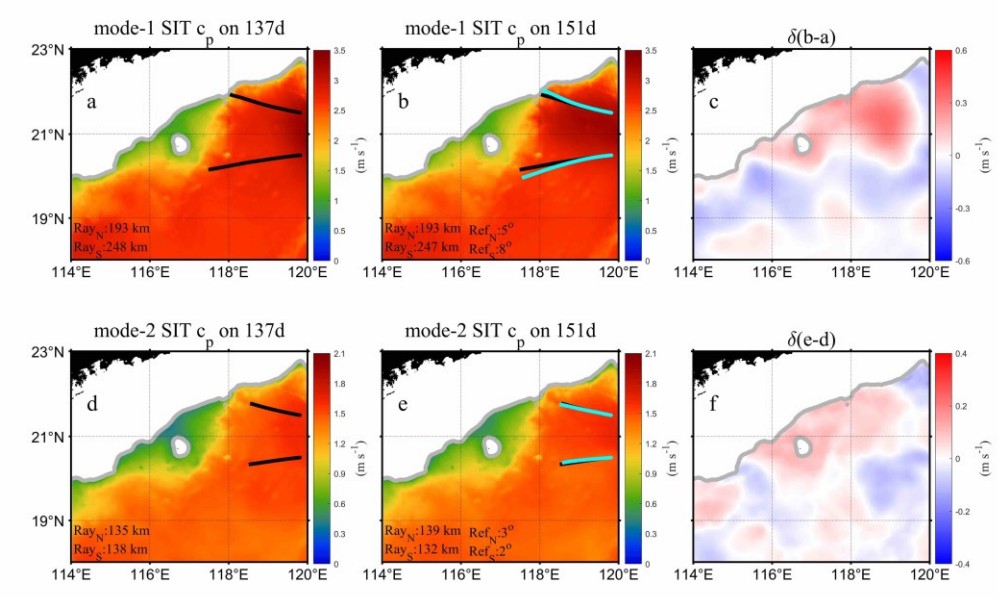

**Figure 13.** Spatial distributions of phase speeds (only considering the stratification) of mode-1 and mode-2 SITs on days 137 and 151. **(c)** and **(f)** represent the differences between days 151 and 137. The black and cyan curves are the propagation rays of SITs during days 137 and 151, respectively.

Figure 14 compares the SIT phase speeds without and with the impact of AE velocity, demonstrating that the northern part of the AE's velocity field can lower the SIT phase speed in the region of (118°-119.9° E, 21.2°-22.3° N), with the mean decreases of modes 1 and 2 being -0.20 and -0.12 m s$^{-1}$, respectively. The SIT phase speed increased in AE's southern part, especially in the region of (118°-119.9° E, 19.9°-21.2° N), with the mean increases of modes 1 and 2 being 0.47 and 0.30 m s$^{-1}$, respectively. The IT rays on day 151 calculated using $c_p$ and $c_p^U$ are shown as black and cyan curves in Fig. 14b and Fig. 14e, from which two significant features can be identified. One is that the propagation distance of the north ray was shortened, while that of the south ray was lengthened. The other effect is that the AE velocity field magnified both the northward deflection of the north ray and the southward deflection of the southern ray. Zhao (2014) found that the propagation of diurnal ITs (e.g., $K_1$ and $O_1$) in the SCS is deflected toward the equator (its phase speed decreases with latitude). Under the influence of the AE, the latitudinal distribution of phase speed was altered (a local maximum occurred on the southern side of the AE and tended to decrease toward both north and south directions), which resulted in the SIT refracted toward both north and south directions, because SIT propagation follows the direction of group velocity (along which phase speed decreases). Finally, the IT rays are shown in Fig. 14b and Fig. 14e, since the acceleration zone is on the south side and the deceleration zone is on the north side of the AE.



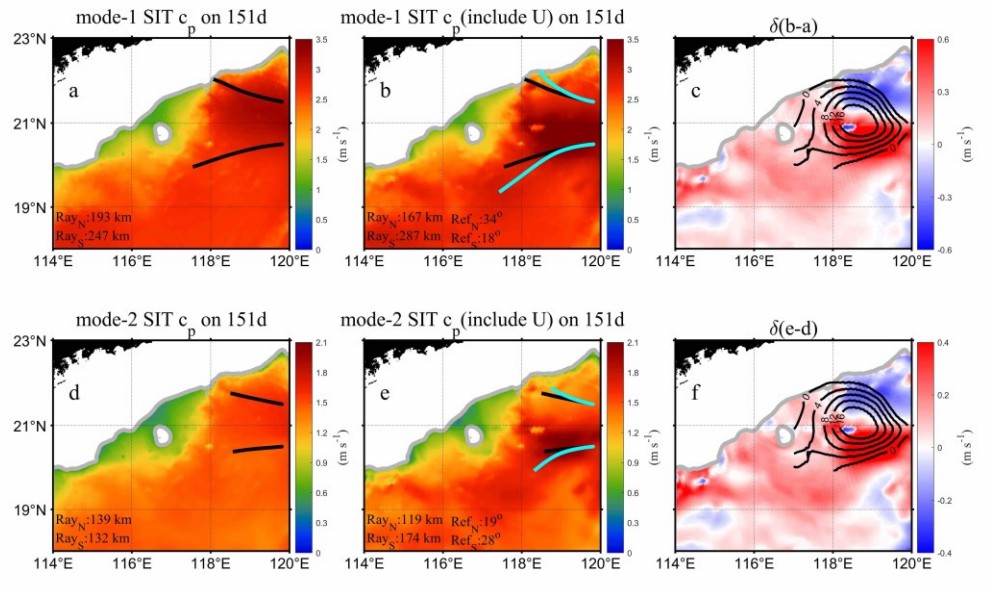

**Figure 14.** Spatial distributions of mode-1 and mode-2 SIT phase speeds on day 151. **(a)** and **(d)** are the same as Fig. 13b and Fig. 13e for $c_p$ without the effect of the background flow; **(b)** and **(e)** are $c_p^U$ considering the effect of the background flow; **(c)** and **(f)** indicate the difference between the first two modes $c_p^U$ and $c_p$; and black contours represent SLA on day 151.

Zaron and Egbert (2014) proposed that the relative perturbations of phase speed ($\delta c_p / c_p$) are mainly composed of three portions, namely, perturbations in stratification, and advection and shear of the background flow. Their results indicated that stratification and advection are the dominant factors for the change of phase speed along the Hawaiian Ridge. The same finding was obtained by Savage et al. (2020) in the Tasman Sea in the southwestern Pacific Ocean. The results of our analysis shown above are consistent with these published ones.

## 4 Discussion

Kelly et al. (2012) introduced the topographic conversion term to quantify the work done by the mode-m IT on mode-n IT, which is related to topography and background stratification (Kelly et al., 2013). The background temperature-salinity field is altered by ME (Hu et al., 2011; Chu et al., 2014; Fernández-Castro et al., 2020). Therefore, the topographic conversion term is one of the contributions of interaction. In addition, the advection and energy exchange terms are directly related to the velocity field of ME. Unlike the energy exchange term, the advection term does not involve energy transfer between ME and IT. In other words, the advection term's energy comes from IT, whereas ME just induces reflection or refraction of IT energy (Zaron and Egbert, 2014; Kelly et al., 2016b; Wang et al., 2021). Next, we discuss their relative importance.

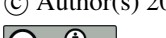



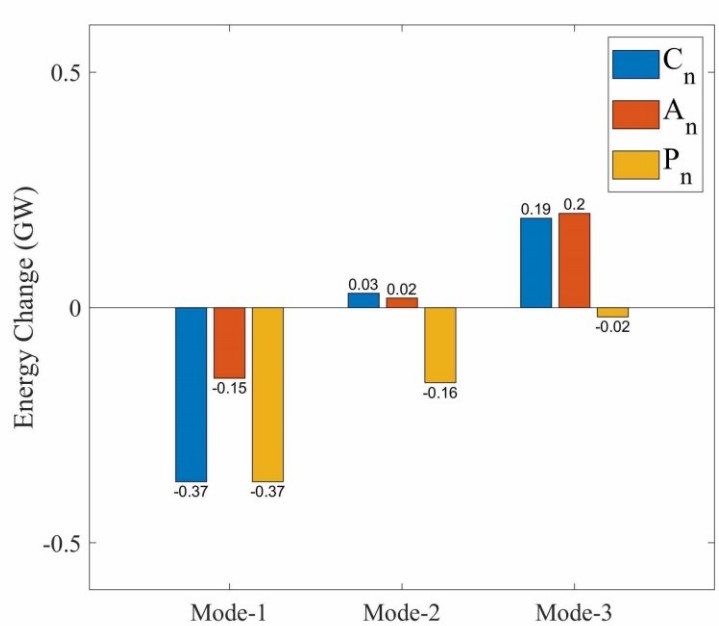

**Figure 15.** Changes in topographic conversion term ($C_n$), advection term ($A_n$), and energy exchange term ($P_n$) for the first three modes on
day 151 compared to those on day 137.

The above results (Fig. 5, Fig. 7, Fig. 8, and Fig. 10) present that among the three main contribution terms, the amplitude
of the advection terms ($A_{mn}^u + A_{mn}^p$) is the largest on day 151, followed by that of the topographic conversion term and energy
exchange term. However, different conclusions can be drawn in terms of their absolute changes on day 151 relative to day
137 (Fig. 15). Although the advection term had the largest amplitude, its absolute change was minor, while a larger change
existed in the topographic conversion and energy exchange terms during this period. Figure 15 illuminates that the advection
and topographic conversion terms had similar changing trends (decreasing in mode 1, increasing in modes 2 and 3), but the
absolute change in the topographic conversion term was significantly larger. Nevertheless, the absolute change in energy
exchange term decreased with the increase of mode number, but it was negative for modes 1-3, indicating energy transfer
from IT to ME. These results suggest that when the AE and SIT interacted near the continental slope, the variation of energy
in the first three modes of SITs was mainly dominated by topographic conversion and energy exchange, rather than by
advection.

## 5 Conclusions

Using a high-resolution dataset of 3D numerical model, we investigated the interaction processes between an AE and SIT on
the continental slope of the northeastern SCS. The main conclusions are as follows.





The interaction between the AE and SIT changed the magnitude of SIT energy. Using the energy equation of IT with ME, we quantified the contributions of three interaction terms. It is evident that the effects of the topographic conversion and energy exchange (including shear production and horizontal buoyancy production) terms on SIT energy far outweighed that of the advection term. Additionally, the AE promoted topographic inner-modal scattering and facilitated downscale cascade

of SIT energy on the continental slope, which drove energy conversion from mode-1 SIT to higher-mode SITs. Moreover, energy was transferred from the IT field to the eddy field at an average rate of -3.0 mW m⁻², mostly via vertical shear production. The advection term weakened mode-1 SIT energy while enhancing mode-2 and mode-3 SIT energy, with velocity-associated advection dominating pressure-related advection.

The interaction not only changed the SIT energy but also modulated the propagation of low-mode SITs. The analysis

based on the Taylor-Goldstein equation confirms that the currents related to the AE have a more significant impact on altering SIT's phase speed than eddy-associated stratification variation. The eddy currents decreased SIT's phase speed on AE's north side, while increasing it on AE's south side. Consequently, the SIT propagation distance is found to be shorter on AE's north side than on the south side within the same time. In addition to causing refraction of low-mode SITs, the interaction may also enhance the reflection of low-mode SITs near the supercritical continental slope. The AE was found to

intensify the energy and propagation distance of the reflected mode-2 SIT, increasing the reflection coefficient from 32% (pre-AE) to 44% (during the AE period).

In summary, the interaction between the AE and SITs altered the energy and propagation of SITs in the SCS, thereby affecting the spatial distribution of tidal-induced turbulent dissipation. Therefore, comprehending the physical mechanism behind their interaction is instructive for parameterization and forecasting of ITs.

Limited by the length of the article, we only analyzed the interaction between the AE and SITs. We plan to investigate the interaction between eddy of other polar (e.g., cyclonic eddy) and (semidiurnal or diurnal) ITs in the future.

**Appendix A**

The normalized eigenfunction from Eq. (7) satisfies the orthogonal condition:

$$\begin{cases} \int_{-H}^{0} \Phi_m \Phi_n N^2 dz = c_n^2 H \delta_{mn}, \\ \int_{-H}^{0} \phi_m \phi_n dz = H \delta_{mn} \end{cases} \tag{A1}$$

where $\delta_{mn}$ is the Kronecker delta (when $m \neq n$, $\delta_{mn}$=0, when $m = n$, $\delta_{mn}$=1).

The formulas for each term in Eq. (1) are as follows:



$$\begin{cases} \boldsymbol{F}_n = H\boldsymbol{u}_n p_n = \boldsymbol{U}_n p_n \\ A_{mn} = A^u_{mn} + A^p_{mn} = [(\overline{\boldsymbol{U}}_{mn} \cdot \nabla)\boldsymbol{U}_m] \cdot \frac{\boldsymbol{U}_n}{H} + (\frac{H}{c_n^2}\overline{\boldsymbol{U}}_{p,mn} \cdot \nabla p_m)p_n \\ C_{mn} = [T_{mn}(\boldsymbol{u}_m p_n) - T_{nm}(\boldsymbol{u}_n p_m)] \cdot \nabla H \\ P^S_{mn} = P^{Sh}_{mn} + P^{Sv}_{mn} = -(\boldsymbol{U}_m \cdot \nabla)\overline{\boldsymbol{U}}_{mn} \cdot \frac{\boldsymbol{U}_n}{H} + (\nabla \cdot \boldsymbol{U}_m)\overline{\boldsymbol{U}}_{z,mn} \cdot \frac{\boldsymbol{U}_n}{H} \\ P^B_{mn} = \left(\frac{\boldsymbol{U}_m}{c_n^2} \cdot \overline{\boldsymbol{B}}_{mn}\right)p_n \end{cases}, \tag{A2}$$

Where $\boldsymbol{u}_n$ and $p_n$ are the mode-n amplitude of baroclinic velocity $\boldsymbol{u}$ and pressure perturbation $p$, respectively. $C_{mn}$ is calculated by using Eq. (12) of Zaron et al. (2022). The operators $\overline{\boldsymbol{U}}_{mn}, \overline{\boldsymbol{U}}_{p,mn}, \overline{\boldsymbol{U}}_{z,mn}, T_{nm},$ and $\overline{\boldsymbol{B}}_{mn}$ are calculated as follows:

$$\begin{cases} \overline{\boldsymbol{U}}_{mn} = \frac{1}{H} \int_{-H}^0 \overline{\boldsymbol{U}} \phi_m \phi_n dz \\ \overline{\boldsymbol{U}}_{p,mn} = \frac{1}{H} \int_{-H}^0 \overline{\boldsymbol{U}} \frac{N^2}{c_m^2} \Phi_m \Phi_n dz \\ \overline{\boldsymbol{U}}_{z,mn} = \frac{1}{H} \int_{-H}^0 \overline{\boldsymbol{U}}(\Phi_m \Phi_n \frac{N^2}{c_n^2} - \phi_m \phi_n)dz \\ T_{nm} = \begin{cases} \frac{1}{2}(1 - \phi_n^2)|_{z=-H}, n = m \\ \frac{c_n^2}{c_m^2 - c_n^2} \phi_n \phi_m|_{z=-H}, n \neq m \end{cases} \\ \overline{\boldsymbol{B}}_{mn} = \begin{cases} \frac{f}{H} \int_{-H}^0 \overline{V} \left(\frac{N^2}{c_m^2}\Phi_m\Phi_n - \phi_m\phi_n\right) dz, \overline{B}_{x,mn} \\ -\frac{f}{H} \int_{-H}^0 \overline{U} \left(\frac{N^2}{c_m^2}\Phi_m\Phi_n - \phi_m\phi_n\right) dz, \overline{B}_{y,mn} \end{cases} \end{cases}, \tag{A3}$$

where $\overline{\boldsymbol{U}}$ is the horizontal velocity of ME, $\overline{\boldsymbol{B}}_{mn}$ is related to ME velocity using thermal wind balance to reduce numerical errors caused by differential calculations (e.g., horizontal and vertical gradient operators) (Book et al., 1975; Lahaye et al., 2020).

**Appendix B**

To evaluate the reasonableness of term $\langle D_n \rangle$ in Eq. (1), we estimated the energy dissipation of the model using the viscosity coefficient of MITgcm LLC4320. According to the official documentation (https://github.com/MITgcm-contrib/llc_hires/tree/master/llc_4320), the horizontal eddy viscosity coefficient is determined by the parameterization scheme, and the vertical eddy viscosity coefficient is $A_v$=5.66×10$^{-4}$ m$^2$ s. The horizontal diffusion coefficient is $K_h$=0 m$^2$ s, and the vertical diffusion coefficient is $K_v$=5.44×10$^{-7}$ m$^2$ s. Because all diffusion coefficients for temperature and salinity are very small, they can be ignored in $\varepsilon_{diff}$ term.

$$D = \varepsilon_{visc} + \varepsilon_{diff} + \varepsilon_{drag}, \tag{B1}$$

$$\varepsilon_{visc} = \rho_c A_h \left[\left(\frac{\partial u}{\partial x}\right)^2 + \left(\frac{\partial u}{\partial y}\right)^2 + \left(\frac{\partial v}{\partial x}\right)^2 + \left(\frac{\partial v}{\partial y}\right)^2\right] + \rho_c A_v \left[\left(\frac{\partial u}{\partial z}\right)^2 + \left(\frac{\partial v}{\partial z}\right)^2\right], \tag{B2}$$



$$\varepsilon_{drag} = \rho_c C_d |\boldsymbol{U}|(Uu + Vv)|_{z=-H} \,, \tag{B3}$$

where $\rho_c$ indicates the reference density, and $A_h$ and $A_v$ are horizontal and vertical eddy viscosity coefficients, respectively. $(u, v)$ and $(U, V)$ are horizontal baroclinic and total velocity, respectively. $C_d$ is the bottom drag coefficient.

To calculate Eq. (B2), we need to know $A_h$. Although the variable is not an output field, we can obtain it through the information of the AE. According to Li et al. (2017), the average eddy viscosity in the SCS is 343 m$^2$ s. Using their method, we estimate that in the AE on day 151, $A_h$=436 m$^2$ s. The result is slightly larger than that in Li et al. (2017). For the

convenience of calculation, we round off the horizontal eddy viscosity to $A_h$=400 m$^2$ s.

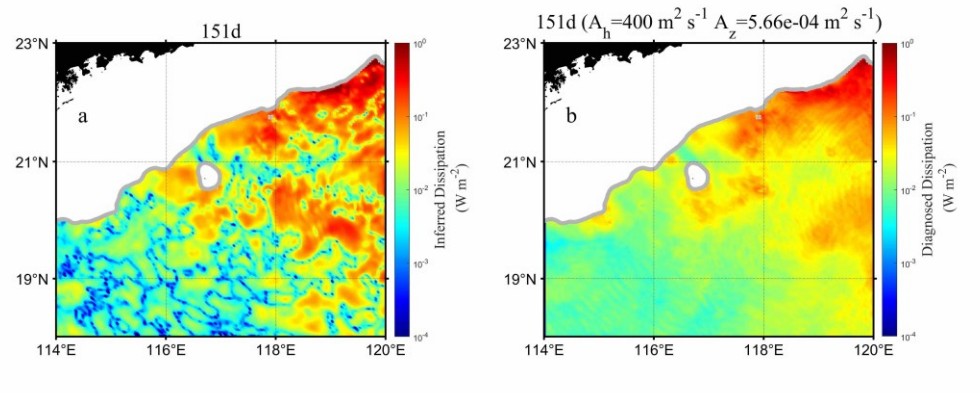

**Figure B1.** Inferred dissipation for the first five modes **(a)** and diagnosed dissipation **(b)** on day 151.

Figure B1a is the inferred result of $\sum_{n=1}^{5}\langle D_n \rangle$ according to Eq. (1), and Fig. B1b is the diagnosed result of Eq. (B1). In terms of magnitude, they both have energy dissipation rates that reach or exceed 10$^{-1}$ W m$^{-2}$ in the northeastern SCS. Their

overall spatial patterns are similar, demonstrating that the energy equation of IT may achieve a better balance when considering the dissipation term. However, there are differences between Fig. B1a and Fig. B1b in some regions (such as near 118° E, 20° N), which may be due to the fact that Fig. B1b uses the constant $A_h$, which is actually determined by the parameterization scheme in LLC4320, and the asymmetry of the eddy field may lead to spatial variations in $A_h$ (Hu et al., 2011; Chen et al., 2012; Xu et al., 2016).

*Data availability.* The MITgcm LLC4320 output is available at https://data.nas.nasa.gov/ecco/data.php, which was provided by Estimating the Circulation and Climate of the Oceans (ECCO) project and supported by High-End Computing (HEC) from the NASA Advanced Superconducting Division at the Ames Research Center. The TPXO-v9 data were obtained from https://www.tpxo.net/global/tpxo9-atlas, provided by Dr. Erofeeva from Oregon State University.

*Author contributions.* The study was conceived and designed by all co-authors. Data preparation, material collection, and
analysis were performed by LF. LF prepared the manuscript with contributions from all co-authors.



*Competing interests.* The contact author has declared that none of the authors has any competing interests.

*Acknowledgments.* We are very grateful to Samuel M. Kelly for the discussions on the energy equation of internal tide. This work is jointly supported by the National Natural Science Foundation of China (grant 42076012, 42376012, 42006012).

*Financial support.* This research has been supported by the National Natural Science Foundation of China (grant no. 42076012, 42376012, 42006012).

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
