# Peer review of "Numerical investigation of interaction between anticyclonic eddy and semidiurnal internal tide in the northeastern South China Sea"

_EGUsphere, 2023_

## Author Comment (AC1)

**RESPONSE TO REVIEWERS**

Note: The reviewer's original comments are indicated in black, and our responses are indicated in blue. Figures in the manuscript are numbered and labeled using Arabic numerals, e.g., Figure 1; those in the response to reviewers file are numbered and labeled using Arabic numerals with a prefix "R", e. g. Figure R1.

**Reviewer #1**

**Summary:**

Based on the MITgcm LLC4320 data, this paper investigates the interaction between semidiurnal internal tide (SIT) and an anticyclonic eddy (AE) in the northern South China Sea. Through calculating the energy budget of the first three modes of SIT, the authors analyze the interaction between the modal SIT and AE. Results indicate that the AE can modulate the intensity and propagation of SIT.

**Comments:**

1. L13, As the authors have pointed out that the energy is transferred from SIT to AE, the value of the transferring rate (-3.0 mW m^-2) should be changed to 3.0 mW m^-2.

**Responses:** Thank you for this suggestion, the negative symbol has been removed from this sentence.

2. L77-79, "The model can effectively simulate free propagating internal waves such as ITs, while regional models cannot because of …". I disagree.

**Responses:** We apologize for using such an improper statement. We are trying to convey that current regional circulation models rarely consider the influence of far-field internal waves (IWs) on the near-field IWs when modeling IWs, because the open boundary condition in the SINGLE RUN case cannot introduce the forcing of low-mode IWs from the external region. A multi-layer nesting strategy, however, can make the regional circulation model include the effect of far-field IWs from the external region. The reviewer is right that the regional circulation model does have such abilities, although its implementation is complex and computationally

demanding. Now we rewrite this sentence as "The model can effectively simulate free-propagating internal waves such as ITs, while regional models get weaker IWs in the simulated region when they do not introduce forcing of low-mode IWs from the external region."

3. L109-110, I do not understand why model decomposition is related to horizontal resolution.

**Responses:** We are sorry for this information gap, the dispersion relation of linear IWs is:

$$k_z = \sqrt{\frac{N^2 - \omega^2}{N^2 - f^2}} k_h \text{ , or } \lambda_h = \sqrt{\frac{N^2 - \omega^2}{N^2 - f^2}} \lambda_z \qquad (R1)$$

for IWs at a fixed frequency $\omega$, the horizontal wavelength $\lambda_h$ is proportion to the vertical wavelength $\lambda_z$, and hence inversely proportional to the mode number. When the horizontal wavelength of the mode-$n$ internal tide is less than two times the horizontal resolution of model ( $\lambda_n^h < 2\Delta x$ ), the mode-$n$ internal tide cannot be identified. As a result, the mode decomposition is related to both the vertical resolution and horizontal resolution.

4. L117 and Figure 1, The regionally averaged barotropic tidal currents do not make sense, because the phases of tidal currents at different points are different.

**Responses:** We agree with the reviewer that the phase of barotropic tide varies spatially. However, from Figure R1, we can see that the mean value of the phase velocity of the semidiurnal barotropic tide in the region R2 is more than 100 m s$^{-1}$ (which can be estimated using $c = \sqrt{gH}$ with a mean water depth of 2000 m), and the corresponding horizontal wavelength is more than 6000 km. The zonal distance of region R2 is about 300 km, meaning that the phase change of semidiurnal barotropic tide could be ignored in region R2.

[Figure]

Figure R1 Spatial distribution of the phase velocity of semidiurnal barotropic tide in the study area, with the three red pentagrams indicating the stations analyzed in Figure R2.

It can be seen clearly from Figure R2 that the phase difference of the zonal velocity of semidiurnal barotropic tide is small, even at different stations. It is reasonable that spatial averaging in a relatively small region does not departure the real results significantly. Moreover, Figure 1b-c are just used for qualitative examination of the spring tidal moment.

[Figure]

Figure R2 Zonal velocity of semidiurnal barotropic tide for the three stations marked with red pentagram in Figure R1. Data are from the TPXO-v9 model, using the TMD toolkit.

5. L135, Please introduce how to calculate the HKE and APE.

**Responses:** We apologize for this gap, the method (Kelly et al., 2012) calculating HKE and APE is:

$$\text{HKE}_n = \frac{\rho_0 H}{2}(u_n^2 + v_n^2) \tag{R2}$$

$$\text{APE}_n = \frac{\rho_0 H}{2}\left[\left(1 - \frac{f^2}{\omega^2}\right)\frac{p_n^2}{c_n^2}\right] \tag{R3}$$

**Reference:**

Kelly, S. M., Nash, J. D., Martini, K. I., Alford, M. H., and Kunze, E. (2012). The cascade of tidal energy from low to high modes on a continental slope. Journal of physical oceanography, 42(7), 1217-1232.

6.  L163-165, The authors use the theoretical estimation of Vic et al. and L_1 of Xu et al. to demonstrate that the simulated mode-2 SIT is consistent with the theory (Figure 3). However, it seems that the simulated mode-1 SIT (Figure 2) is not consistent with L_1 of Xu et al. Moreover, it seems that the calculated L_3 (L178) is not consistent with the result shown in Figure 4.

**Responses:** We are sorry for this confusion, the estimates of propagation distance based only on empirical equation may have a bias, instead, we calculated the SIT's energy fluxes directly throughout the whole SCS (at depths greater than 200 m) using LLC4320 output. Take day 151 as an example in Figure R3. The propagation distance of mode-1 SIT exceeds 1000 km, which is consistent with the $L_1$ of Xu et al. (2016). The propagation distance of mode-2 SIT is about 400 km, which is larger than the result of empirical equation (125-188 km). Similarly, the propagation distance of the mode-3 SIT is about 80 km, also larger than the result of empirical equation. We acknowledge that the empirical equation $L_n \approx L_1/n^3$ can reflect the propagation distance decreasing with the increasing mode number qualitatively, but its quantitative estimation of the propagation distance is less precise.

[Figure]

Figure R3 Spatial distribution of energy flux for the first three modes of SITs in the SCS.

**Reference:**

Xu, Z., Liu, K., Yin, B., Zhao, Z., Wang, Y., and Li, Q. (2016). Long-range propagation and associated variability of internal tides in the South China Sea. Journal of Geophysical Research: Oceans, 121(11), 82688286.

7. Figure 6. If the IT is locally generated, the corresponding conversion from barotropic tides ($C\_0x$, x=1,2,3,…) should be positive. If the local IT is influenced by that propagated from remote source, the value of conversion might be negative. It is generally recognized that high-mode IT cannot propagate a long distance from the source. To be specific, high-mode IT (especially mode-4 and mode-5) generated at the Luzon Strait might not reach the study region. Therefore, how to explain the negative values of $C\_04$ and $C\_05$?

**Responses:** we thank the reviewer for this valuable comment. First, $C_{0n}$ is examined by comparing with depth-integrated energy conversion from barotropic tide to all baroclinic modes $E_{bt2bc}$ (calculated by Equation R5). It is found that the spatial distribution of $\sum_{n=1}^{5} C_{0n}$ (Figure R4a) is nearly the same as that of $E_{bt2bc}$ (Figure R4b), which indicates a reliable calculating of $C_{0n}$.

$$C_{0n} = -\nabla H \cdot \boldsymbol{u_0}(p_n\phi_n)|_{z=-H} \qquad (\text{R4})$$

$$E_{bt2bc} = g \int_{-H}^{\eta} \rho' w_{bt} dz \qquad (\text{R5})$$

[Figure]

Figure R4 (a-b) Spatial distribution of barotropic to baroclinic energy conversion rates on day 151 in the northern SCS, calculated using Equations R4 and R5, respectively.

Second, the positive or negative of the barotropic to baroclinic energy conversion rate in Equation R4 is determined by topographic gradient, barotropic velocity, and bottom pressure perturbation (Zilberman et al., 2009), i.e., the meridional component of $C_{0n}^{y}$:

$$C_{0n}^{y} = -\frac{dH}{dy}\hat{p}_n\hat{v}_{bt}cos(\theta_p - \theta_v) \qquad (R6)$$

where $\hat{p}_n$ and $\theta_p$ are the amplitude and Greenwich phase of bottom pressure perturbation for $n$th mode, and $\hat{v}_{bt}$ and $\theta_v$ are the amplitude and Greenwich phase of meridional barotropic velocity, respectively. We calculated the conversion rate from barotropic tide to mode-4 baroclinic tide $C_{04}$ (for the northern region of R2 only, where negative values occur) using Equation R6. Figure R5 shows that $C_{04}$ is mainly regulated by its meridional component $C_{04}^{y}$, since the meridional topographic gradient is essentially positive at depths from 250 to 2000 m, the positive or negative of conversion rate is shaped by the cosine of the phase difference $(\Delta\theta = \theta_p - \theta_v)$, which is negative near the 250-500 m isobath at 22°N (Figure R5d), resulting in that $C_{04}$ is predominantly negative in the northern region of R2.

[Figure]

Figure R5 (a-c) Spatial distribution of the conversion rate from barotropic tide to mode-4 baroclinic tide $C_{04}$, as well as its zonal component $C_{04}^x$ and meridional component $C_{04}^y$ on day 151, (d) cosine of the phase difference between bottom pressure perturbation and meridional barotropic velocity, and (e) meridional topographic gradient with three isobaths of 250 m, 500 m, and 2000 m, respectively.

At last, we examine the northward energy fluxes of the first five modes of SITs in the region R2 (Figure R6), and find that when the mode-1 SIT propagates toward the northern SCS (Figure R6a), mode 1 transmits higher-mode SITs over subcritical continental slope (Figure R6b-e), i.e., the region between the magenta contour and the 2000 m isobath. Wang et al. (2018) demonstrate that the subcritical shelves are more conducive to the occurrence of transmission of higher-mode internal tides. Based on the direction of energy fluxes (Figure R6d-e), it can be seen that the transmitted mode-4 and mode-5 SITs propagate northward, whereas the locally-generated mode-4 and mode-5 SITs propagate southward, which are in the opposite directions, leading to the negative values of $C_{04}$ and $C_{05}$ in this region. Negative values of the conversion rate indicate that the internal tide energy is transferred to the barotropic tide through pressure work, which is not involved in turbulent kinetic energy dissipation. In addition, negative conversion rates have been seen in some studies, e.g., Figure 5 of

Song and Chen (2020), Figure 6 of Wang et al. (2016), and Figure 9 of Xu et al. (2016).

[Figure]

Figure R6 Spatial distribution of the northward component (incident wave) of the energy flux (arrows) for the first five SITs modes on day 151, superimposed on the contour map of topographic steepness parameters from 250 to 2000 m, with magenta contour for $\gamma = 1$.

**References:**

Song, P., and Chen, X. (2020). Investigation of the internal tides in the Northwest Pacific Ocean considering the background circulation and stratification. Journal of Physical Oceanography, 50(11), 3165-3188.

Wang, S., Chen, X., Li, Q., Wang, J., Meng, J., and Zhao, M. (2018). Scattering of low-mode internal tides at different shaped continental shelves. Continental Shelf Research, 169, 17-24.

Wang, X., Peng, S., Liu, Z., Huang, R. X., Qian, Y. K., and Li, Y. (2016). Tidal mixing in the South China Sea: An estimate based on the internal tide energetics. Journal of Physical Oceanography, 46(1), 107-124.

Xu, Z., Liu, K., Yin, B., Zhao, Z., Wang, Y., and Li, Q. (2016). Long-range propagation and associated variability of internal tides in the South China Sea. Journal of Geophysical Research: Oceans, 121(11), 82688286.

Zilberman, N. V., Becker, J. M., Merrifield, M. A., and Carter, G. S. (2009). Model estimates of $M_2$ internal tide generation over Mid-Atlantic Ridge topography. Journal of Physical Oceanography, 39(10), 2635-2651.

8.  Section 3.2.1 and corresponding content in section 3.1.1, The authors find that the calculated r_E is different from the theoretical r_E for mode-2 SIT and speculate that it is caused by the interference of SIT after a reflection at the continental slope. Hence, they analyze the reflection of mode-2 SIT in section 3.2.1. I have several questions for this analysis. First, if mode-2 SIT is reflected at the slope, mode-1 and mode-3 SITs are also reflected at the slope. Why only mode-2 SIT causes interference as well as a r_E different from the theoretical value? Second, as shown in Figure 11, the incoming and reflected energy fluxes are in different directions, how to form interference in this case?

**Responses:** we thank the reviewer for this valuable comment. We decompose the incident and reflected energy fluxes of mode-1 and mode-3 SITs (Figure R7 and Figure R8). Figure R7 and Figure R8 share a similar pattern for mode 1 and mode 3, indicating that energy reflection occurs when the incident SITs encounter the continental slope, which confirms that topographic reflection exists in different modes. However, it is difficult to directly judge the presence or absence of reflection from the direction of mode-1 energy flux, which is not referred to in the manuscript.

There are two reasons why the $r_E$ is different from theoretical value. One is topographic reflection of lower mode SITs, the other is interference between transmitted (onshore, Figure R6c) and locally-generated (offshore) higher mode SITs. The combined effect of reflection and transmission leads to a deviation from the theoretical values of $r_E$ for the first three modes, with the absolute errors being 9%, 16%, and 18% on day 151.

[Figure]

Figure R7 The energy flux of mode-1 SIT on day 151 in (a), (b), and (c) for total, northward, and southward, respectively. The energy fluxes integrated along sections S1 and S2 are labelled as onshore and offshore values, respectively. The topographic steepness parameter for SIT on day 151 is presented in (d).

[Figure]

Figure R8 Same as Figure R7, but for the mode 3.

9. Figure 11a, Compared with previous studies (e.g. Kerry et al., 2013; Xu et al., 2021), the energy flux pattern of SIT shown in this study is odd.

**Responses:** We are sorry for this confusion. For the energy flux of mode-1 SIT in Figure R7a, it is generally in agreement with those of the earlier studies, e.g., Zhao (2014). The energy flux bifurcates around (119.5°E, 21.0°N) when the Kuroshio loop is present, as indicated in both Kerry et al. (2013) and Xu et al. (2021)'s work. The incoming energy fluxes at the continental slope reported by Kerry et al. (2013) and Xu et al. (2021) are the total value of all modes, with magnitudes up to 40 kW m$^{-1}$ and 25 kW m$^{-1}$, respectively. Our result of incoming energy fluxes of mode-2 SIT is 15 kW m$^{-1}$, among which the maximum of the reflected part is 8 kW m$^{-1}$, in this way, the reflection of mode-2 SIT is difficult to observe in Kerry et al. (2013) and Xu et al. (2021)'s results.

**References:**

Kerry, C. G., Powell, B. S., and Carter, G. S. (2013). Effects of remote generation sites on model estimates of M$_2$ internal tides in the Philippine Sea. Journal of Physical Oceanography, 43(1), 187-204.

Xu, Z., Wang, Y., Liu, Z., McWilliams, J. C., and Gan, J. (2021). Insight into the dynamics of the radiating internal tide associated with the Kuroshio Current. Journal of Geophysical Research: Oceans, 126(6), e2020JC017018.

Zhao, Z. (2014). Internal tide radiation from the Luzon Strait. Journal of Geophysical Research: Oceans, 119(8), 5434-5448.

10. L322-326, The authors find that the reflected mode-2 SIT on day 151 is larger than that on day 137, and then conclude that the AE promotes the reflection of SIT. This is imprecise, because the incoming SIT is also increased from day 137 to 151 (section 3.1.1).

**Responses:** We thank the reviewer for this valuable comment. As the reviewer pointed out, the incident mode-2 SIT on day 151 (0.70 GW) is larger than that on day 137 (0.31 GW), and more internal tide energy is reflected on day 151 (0.31 GW) than

on day 137 (0.10 GW). Meanwhile, we compare the reflection coefficient (i.e., the reflected energy divided by the incident energy), which is 32% on day 137 and increases to 44% on day 151, increasing by 12% under the influence of an anticyclonic eddy (on day 151), implying that the eddy promotes a reflection of mode-2 SIT. We added this in the manuscript.

11. L347, There is no c^U in Equation (6).

**Responses:** We correct this spelling error. It should be $c_n^U$.

12. L350, There is no c in Equation (7).

**Responses:** We correct this spelling error. It should be $c_n$.

---

## Author Comment (AC2)

**RESPONSE TO REVIEWERS**

Note: The reviewer's original comments are indicated in black, and our responses are indicated in blue. Figures in the manuscript are numbered and labeled using Arabic numerals, e.g., Figure 1; those in the response to reviewers file are numbered and labeled using Arabic numerals with a prefix "R", e. g. Figure R1.

**Reviewer #2**

**Summary and Recommendation:**

This study examines data from high resolution numerical simulations, focusing on the northeastern South China Sea which experiences internal tides propagating from their generation site at the Luzon Straits. A short period of just under 3 spring-neap tidal cycles is considered, and the internal tide energy fluxes are compared between two periods 2 weeks apart, one of which includes an anticyclonic mesoscale eddy, while the other has no eddy. This comparison reveals that there is a net transfer of energy from the internal tides to the eddy (at least when only the first 3 modes are considered) and that the eddy facilitates topographic conversion from mode 1 to higher modes. The role of the eddy in refracting the internal tide energy flux is also considered, and in affecting changes in reflection at the continental slope. Overall, while this manuscript examines only one instance of internal tide/mesoscale eddy interaction, there are nonetheless many interesting results. However, I recommend some more effort to explain the causes of some of the behavior, and more connection made between the different changes identified.

**Responses:** We thank the reviewer very much for his/her constructive comments. These comments are valuable and helpful for improving our manuscript, and accordingly we have made extensive revisions to our manuscript to make our results convincing. Detailed corrections are listed below.

**Significant suggestions:**

1.  Energy transfer from internal tide to eddy, v. inter-modal energy transfer.

From eqn 1, the authors have identified a net energy transfer from the internal tide to the eddy. However, it would be of interest to know whether this is manifest in an

increase in eddy energy, or whether there a subsequent energy transfer from the eddy to higher internal tide modes. Can you separately diagnose the net mode-mode transfer, as in https://doi.org/10.1175/JPO-D-23-0045.1? Or can you track the eddy energy tendency - does the eddy energy in fact increase due to the internal tide energy transfer?

**Responses:** We understand the reviewer's concern. We examine the eddy kinetic energy (EKE) and its tendency along the eddy trajectory in Figure R1:

[Figure]

Figure R1 (a) The time variation of EKE, the red squares represent the days 137, 151, and 164. (b) The black line represents the tendency of EKE, and the blue line represents the change in shear production term (Ps). Note that negative Ps indicates energy transfer from internal wave field to eddy field.

Figure R1a shows that the EKE increases firstly and then decreases, with its maximum value on day 151. This indicates that before day 151, the eddy continuously gains energy from other motions, and then gradually loses energy and dissipates afterwards. From Figure R1b, we can also find that the tendency of the EKE is positive before day 151, corresponding to its energy growth phase. Afterward, it becomes mainly negative, corresponding to its energy decay phase. During the whole period (40 days), most values of the shear production term (Ps) are negative, indicating that internal tides continuously transfer energy to the eddy field. It seems

like there being a contradiction that both Ps and dEKE/dt are negative during the last 20 days. We think this may be due to other dynamic motions in the northern SCS such as Kuroshio intrusion, submesoscale motions and internal waves at various frequency bands, which contribute to variations in EKE along with SIT-to-eddy conversion discussed here accounting for a fraction of total energetics. It was also reported that EKE tendency in the northern SCS could involve advection effect by large-scale mean flow as well as barotropic/baroclinic instability processes (Liu et al., 2022, Figure R2).

[Figure]

Figure R2 Cited from Liu et al. (2022). The tendency of EKE (Et), the advection of EKE by large-scale circulation (ADV), the perturbation pressure work divergence (PD), the energy conversion between EKE and eddy available potential energy through baroclinic instability (BC), the energy conversion between large-scale and mesoscale kinetic energy through barotropic instability ($BT_{LM}$), the energy conversion between mesoscale and s mesoscale kinetic energy through barotropic instability ($BT_{SM}$ term), the interaction between mesoscale and submesoscale processes (RS term), the EKE dissipation caused by horizontal eddy viscosity (Dah term), the wind stress work (WW term), the friction work at the sea bottom (WB term).

**Reference:**

Liu, Y., Zhang, X., Sun, Z., Zhang, Z., Sasaki, H., Zhao, W., and Tian, J. (2022). Region-dependent eddy kinetic energy budget in the northeastern South China Sea revealed by submesoscale-permitting simulations. Journal of Marine Systems, 235, 103797.

2.  Energy flux arrows

It would be very helpful to see arrows showing the energy flux in the figures 2a-c, 3a-c, and 4a-c. Which direction is the internal tide energy coming from? For example, is mode 1 predominantly coming from the Luzon Straits, while mode 3 is coming from the local continental slope, due to the topographic conversion from mode 1 to 3?

**Responses:** We understand the reviewer's concern. As the reviewer suggested, we add arrows showing the SITs' energy fluxes to Figures 2-4. The Figures R3-R5 show that mode-1 SITs are generated from the Luzon Strait and propagate toward the northern SCS, the mode-2 SITs are similar to mode 1, which propagates mainly westward from the Luzon Strait. However, mode 3 is generated mainly near the continental slope in the northern SCS and propagates southward.

[Figure]

Figure R3 (a-c) Spatial distribution of mode-1 SIT energy on days 137, 151, and 164. Black

arrows represent energy flux, and grey contours represent the depth of 250 m. (d-f) Time series of TE, HKE, and APE obtained from area integral over the region R2, with the red diamonds corresponding to days 137, 151, and 164, respectively. The grey curve in (f) is calculated using Eq. (2).

[Figure]

Figure R4 Same as Figure R3, but for mode-2 SIT.

Figure R5 Same as Figure R3, but for mode-3 SIT.

3. How does the presence of the eddy contribute to the enhanced topographic scattering from mode 1 to higher modes?

Can we connect the enhanced topographic scattering in figures 5 and 6 to the influence of the eddy on the mode 1 propagation shown in figure 14? Does the redirection of the mode 1 toward the slope lead to the increase topographic energy conversion from mode 1 to higher modes?

**Responses:** We thank the reviewer for this valuable comment, which gives us ideas to explore the modal enhancement of the higher-mode SITs. The onshore energy flux for the first five modes crossing the red line in Figure R6 and Figure R7 are calculated by integrating flux values along the section (red line). For mode 1, the onshore flux is 3.19 GW and the offshore value is 1.06 GW on day 137, resulting in a reflection of 33%. On day 151, the onshore energy flux is 5.22 GW and the offshore value is 1.52 GW, with a reflection of 29%, decreasing by 4%. For mode 2, the onshore energy flux is 0.31 GW and offshore value is 0.10 GW on day 137, with a reflection of 32%. On day 151, the onshore energy flux is 0.70 GW and off shore value is 0.31 GW, with a reflection of 44%. The onshore energy flux for modes 3-5 on day 137 are 0.03 GW, 0.01 GW, and 0.01 GW, respectively, while the onshore values on day 151 changed to 0.08 GW, 0.02 GW, and 0.01 GW. As a result, we inferred that the increased higher modal SIT energy flux on the continental slope came from the mode-1 SIT (mode-1 SIT had a reduced reflection on day 151), which may be due to transmission of the mode-1 SIT as it passes the subcritical continental slope, transferring energy from the lower mode to the higher modes. It can be checked by Figure R7 that modes 3-5 have the remarkable energy flux vectors between the critical topography ($\gamma = 1$ magenta curve) and the 2000 m isobath. It is also reported that the low-modal internal tides passing through the subcritical continental slope topography are more susceptible to transmission (Hall et al, 2013; Wang et al., 2018 and 2019), consistent with our results.

[Figure]

Figure R6 Spatial distribution of the northward component (incident wave) of the energy flux (arrows) for the first five SITs modes on day 137, superimposed on the contour map of topographic steepness parameters from 250 to 2000 m, with magenta contour for $\gamma = 1$. The red values in the upper left are calculated by integrating energy flux along the section (red line).

[Figure]

Figure R7 Same as Figure R6, but for day 151.

**References:**

Hall, R. A., Huthnance, J. M., and Williams, R. G. (2013). Internal wave reflection on shelf slopes with depth-varying stratification. Journal of Physical Oceanography, 43(2), 248-258.

Wang, S., Chen, X., Li, Q., Wang, J., Meng, J., and Zhao, M. (2018). Scattering of low-mode internal tides at different shaped continental shelves. Continental Shelf Research, 169, 17-24.

Wang, S., Chen, X., Wang, J., Li, Q., Meng, J., and Xu Y. (2019). Scattering of low-mode internal tides at a continental shelf. Journal of Physical Oceanography, 49(2), 453-468.

4.  Reflection at continental slope - mode 1

In section 3.2.1 the impact of the eddy on the reflection of mode 2 is examined. However, mode 1 is not mentioned here - why not? It would be interesting to know whether the increased topographic conversion from mode 1 to higher modes in the presence of the eddy also leads to reduced reflection of mode 1.

**Responses:** We thank the reviewer for the valuable suggestion. As the reviewer suggested, we applied the decomposition method of incident and reflected waves to mode-1 SIT and got the map of energy flux in Figure R8. This figure shows that the energy reflection also occurs in the first mode as the mode 1 SIT passes over the continental slope. The reflection of mode 1 influenced by eddy on day 151 is slightly smaller than that without eddy influence on day 137, implying that mesoscale eddy contributes to the reduced reflection of mode 1 SIT. Meanwhile, this suggests that the energy of mode-1 SIT is partially involved in the generation of the higher-mode SITs.

[Figure]

Figure R8 The energy flux of mode-1 SIT on day 151 in (a), (b), and (c) for total, northward, and southward, respectively. The energy fluxes integrated along sections S1 and S2 are labelled as onshore and offshore values, respectively. The topographic steepness parameter for SIT on day 151 is presented in (d).

5.  Reasons for enhanced reflection of mode 2

While the authors have shown that the eddy leads to enhanced reflection of mode 2 at the slope, I don't see much explanation of this change - how does the eddy influence this enhanced reflection? Is it due to the refraction of the internal tide toward the slope? Or is it due to the changes in stratification structure induced by the eddy influencing the slope criticality?

**Responses:** We are sorry for this information gap. We show the rose diagram and topographic steepness parameter of the mode-2 SIT energy flux in Figure R9.

[Figure]

Figure R9 (a-b) The rose diagrams for the mode-2 SIT on days 137 and 151, respectively, the selected calculated region is (119-120° E, 19.9-22.3° N). (c-d) The topographic steepness parameter for days 137 and 151, respectively, magenta line for γ = 1 and cyan line for the 2000 m isobath.

From Figure R9(a-b), it can be clearly seen that the eddy has a significant impact on the propagation direction of the mode-2 SIT. On day 137, the mode-2 SIT mainly propagates toward west (180°) and west-northwest (157.5°). It is generally parallel to the critical topography (magenta curve) with a nearly east-west orientation around 119°E. Therefore, the topographic reflection was suppressed due to the angle of the incident waves on day 137. In contrast, on day 151, the mode-2 SIT is deflected by eddy towards the continental slope, and its propagation direction changes to northwest (135°) and west-northwest (157.5°). The angle between its propagation direction and critical topography (magenta curve) increases, which facilitates the reflection of mode-2 SIT. By comparing the topographic steepness parameters on days 137 and

151 (Figure R9c-d), we find that their differences are slight (the magenta curves in these two panels are almost identical), suggesting that the increase in reflection is due to the change of incident angle caused by refraction of the mode-2 SIT due to eddy. We added the related text to the revision.

**Minor comments:**

6. Abstract, line 13-14: The rate at which energy is transferred from the internal tide to the eddy is given here. To know how significant this is, what is this transfer rate as a percentage of the incoming energy flux integrated over the eddy diameter/height?

**Responses:** We understand the reviewer's concern. Seen from Figure R3b-R5b, it is found that the internal tidal energy of the first three modes in the region R2 is 4.92 GW, and the energy transferred from SIT to eddy in this region is approximately 0.33 GW, leading a transfer rate of 7%. However, the SIT in the region R2 varies with time significantly. For example, the internal tidal energy of first three modes is 3.65 GW on day 137, we worried that using a transfer rate may not be universal.

7. Introduction, line 25: delete "and so on", since it does not provide any additional information.

**Responses:** We thank the reviewer. We made this revision.

8. Line 33: More correctly, it is not the dissipation which affects the overturning circulation, but rather the mixing that may be induced by the loss of energy from internal tides.

**Responses:** We are sorry for this confusion. Yes, the circulation could be shaped by the mixing which could transform the water masses and modulate density distribution. In the revision, we made this correction.

9. Line 37: "a hotspot" - a hotspot of what?

**Responses:** We would like to refer to a hotspot of studying multiscale dynamical motions.

10. Line 41-42: If transfer of energy from the mesoscale eddy to the internal wave field induces a viscous effect, this would be a viscous effect on the eddy circulation, not on the internal wave field (which increases in energy in this statement). So it's doubtful that an eddy viscosity can be used to parameterize this effect in an internal tide prediction model.

**Responses:** The reviewer is right. We removed this paragraph for improving the readability of the whole manuscript.

11. Line 51 and elsewhere: Change "inner-modal redistribution" to "inter-modal redistribution" (it is a redistribution between modes).

**Responses:** We understand the reviewer's concern and make this revision.

12. Figure 7, caption: Change "but for the advection term of the velocity component.." to "but for the velocity component of the advection term..."

**Responses:** We thank the reviewer. We made this revision.

13. Figure 8, caption: Change "but for the advection term of the pressure component.." to "but for the pressure component of the advection term.."

**Responses:** We thank the reviewer. We made this revision.

14. Lines 317-325: There is some repetition here. For example line 322-323 closely repeats line 317-318.

**Responses:** We thank the reviewer. We removed repetitions and merged some sentences in the revised manuscript.

15. Line 334-335, and elsewhere: "near-filed" and "far-filed" should be "near-field" and "far-field".

**Responses:** We are sorry for this typo, the word has been corrected.

16. P18, lines 362-368: Too much space is given here to showing that the stratification changes alone have little impact on the ray propagation. I think you could combine figures 13 and 14 and focus on discussing the more significant impact of the eddy flow field.

**Responses:** We understand the reviewer's concern. As the reviewer suggested, the manuscript was revised with focusing on the analysis of eddy flow field.

---

## Author Response (AR2)

**RESPONSE TO REVIEWERS**

Note: The reviewer's original comments are indicated in black, and our responses are indicated in blue. Our changes in the marked-up manuscript version are given in green. Figures in the manuscript are numbered and labeled using Arabic numerals, e.g., Figure 1; those in the response to reviewers file are numbered and labeled using Arabic numerals with a prefix "R", e. g. Figure R1.

**Reviewer #3**

**Comments:**

1. Lines 14-15: In the Abstract, the authors say that "the AE can modify the spatial distribution of tidal-induced dissipation by both refracting and reflecting low-mode SIT"; In lines 49-50, they say that "ME mostly modulates the propagation of IT in terms of refraction and scattering". What is more important, reflecting or scattering?

**Responses:** We are sorry for this confusion. As far as we know, the scattering mechanism is more significant, because internal tides can scatter energy over both steep continental slopes and flat abyssal plains, due to variations in stratification. However, the reflection of internal tides primarily occurs in areas of large topographic gradients (supercritical topography), usually with a slope steepness parameter exceeding 1. In a spatial sense, the impact of internal tide scattering is more widespread than that of internal tide reflecting in terms of occurrence probability. Now we rewrite the sentence in Lines 14-15 as "the AE can modify the spatial distribution of tidal-induced dissipation by refracting, scattering, and reflecting low-mode SIT."

(Change is made in Line 15 in the marked-up manuscript version.)

2. Lines 38-39: The authors say that the internal tide and mesoscale eddy have "comparable horizontal scales", and then they use the "multiscale" to describe their interaction. There seems to be some inconsistency in the phrases "comparable scales" and "multiscale".

**Responses:** We apologize for this unclarity. The horizontal scale of low-mode internal tide is comparable to that of mesoscale eddy; however, the horizontal scale of higher-mode internal tide is much smaller than that of mesoscale eddy. The interaction of mesoscale eddy with internal tide involves multi-modal (namely multiscale) internal tides (e.g., through the scattering process), so we describe it using the word "multiscale." Now we rewrite this sentence as "Due to the comparable horizontal scales of low-mode IT and ME, their interaction occurs easily and becomes a hotspot for studying multiscale dynamical motions

(interaction process involves multi-modal/multiscale internal tides).”

(Changes are made in Lines 37–38 in the marked-up manuscript version.)

3. Line 69: "AE" and "SIT" have been defined in the abstract, but they should be defined again in the text. They are a bit abrupt here in line 69.

**Responses:** We added relevant definitions in the main text.

(Changes are made in Lines 67–68 in the marked-up manuscript version.)

4. Line 114: There's a small error in the formula of the dispersion relation of linear IWs. Sqrt((N^2-ω^2)/(N^2-f^2)) should be Sqrt((N^2-ω^2)/(ω^2-f^2)).

**Responses:** We have corrected this typo and examined other formulas.

(Change is made in Line 113 in the marked-up manuscript version.)

5. Line 117: The authors say "We selected a period for analysis, corresponding to 131-170 days". When using TPXO, the data necessarily corresponds to definite dates. Why don't you use the date here corresponding to the TPXO, instead of the number of days in the model? It feels like this would lead to a lot of inconvenience in the author's following description.

**Responses:** We understand the reviewer's concern. Based on two considerations, we chose to use the days of the model rather than real dates. First, the model days are expressed more briefly, which makes it easier to label as many moments as possible in the drawing; second, because the number of days in each month is not a constant, it is not as intuitive as Arabic numerals in judging unlabeled moments. Meanwhile, some observations (e.g., Osborne et al., 2011; Xie et al., 2013) are also labeled using "yearday."

**References:**

Osborne, J. J., Kurapov, A. L., Egbert, G. D., and Kosro, P. M. (2011). Spatial and temporal variability of the $M_2$ internal tide generation and propagation on the Oregon shelf. Journal of Physical Oceanography, 41(11), 2037-2062.

Xie, X., Shang, X., van Haren, H., and Chen, G. (2013). Observations of enhanced nonlinear instability in the surface reflection of internal tides. Geophysical Research Letters, 40(8), 1580-1586.

6. Line 135: the author do not seem to mention what "red curve (in Figure 2)" means.

**Responses:** We apologize for this information gap. The red curves are the main propagation path for different modes of semidiurnal internal tide. We have added this information in the revised version.

(Changes are made in Lines 136–137 in the marked-up manuscript version.)

7. Line 168 (Figure 3d-f): The Moon's orbit around the Earth is elliptical, with a change in perigee and apogee, a period of about 27 days. The interval between the three spring tides described in this article is exactly 27 days (137 to 164), which makes one wonder if the change in the energy of the internal tides is related to this 27-day inequality of tide. The data used by the authors are model data and should not include this factor of variation in the equinoctial tide. However, the authors should check the model and rule out this possibility.

**Responses:** We thank the reviewer for this valuable comment. By carefully inspecting the model configuration, we confirm that the tidal forcing of the LLC4320 is a body forcing implementation, which is called tidal geopotential forcing in the model. Tidal geopotential forcing considers the celestial gravitational tidal force between the Sun, the Moon, and the Earth, which is described in https://github.com/joernc/tidal-potential. In our view, the model output contains the variation of the equinoctial tide. However, the magnitude of barotropic tide varies only by 10% between the first two spring tide moments (Figure 1b), which is about 1/3 of the variation in the energy of the semidiurnal internal tide, so we believe that the energy of semidiurnal internal tide is mainly influenced by the anticyclonic eddy.

(Changes are made in Lines 123–124 in the marked-up manuscript version.)

8. Line 248: I did not find explanation on mode 0 in Figure 7. Is it the barotropic tide, or AE?

**Responses:** Mode 0 is the barotropic tide. We added this explanation in the revised version.

(Change is made in Lines 251 in the marked-up manuscript version.)

9. Line 251: The word "respectively" seems unnecessary.

**Responses:** We removed this word.

(Change is made in Line 251 in the marked-up manuscript version.)

10. Lines 268, 535: A "*" indicates ordinary multiplication in computer language, but not in math

language. When a "*" is occasionally used in math to denote multiplication, it must be a special defined multiplication, such as convolution. Likewise, expressing powers of 10 in terms of "e" (Line 535, Figure B1) is also not normal in math language. "e" equals 2.71828... in math.

**Responses:** The reviewer is right, we apologize for these improper uses of "*" and "5.66e-4", and we have replaced them with "×" and "5.66×10$^{-4}$".

(Changes are made in Line 269 and Figure B1 in the marked-up manuscript version.)

11. Line 359 (Figure 14b-c): The authors use "onshore (Northward)" and "offshore (Southward)" to present the direction of energy fluxes integrated along section S1 and S2. In my view, section S1 is along-shore and the energy integrated along it is cross-shore; section S2 is cross-shore and the energy integrated along it is along-shore.

**Responses:** The reviewer is right. In the original manuscript, we use the words "onshore" and "offshore" to describe the behavior of mode-2 semidiurnal internal tides when they encounter the continental slope. The incident waves (toward the shore) are reflected, and the reflected waves propagate in a direction away from the shore. As the reviewer suggested, we changed "onshore" to "cross-shore" and "offshore" to "along-shore" to depict this process accurately in the revised text.

(Changes are made in Lines 261-267, 364, 550, Figures 14 and C1-C2 in the marked-up manuscript version.)

12. Line 374 (Figure 15): Typically, wave rays can be found in the contours of the current speed. But here the contours of the current speed show no clear pattern of the wave rays. I wonder how do the author give the black wave rays in Figure 15?

**Responses:** We understand the reviewer's concern. We have only filled in the baroclinic velocity of mode-2 SIT in Figure 15. Examining closely, we can see that the black wave ray passes mainly through regions with positive velocity. Assuming that mode-2 SIT propagates mainly from the continental slope where the topographic steepness parameter is greater than 1, and then integrating the equation (R1) over time gives us the black wave ray in the figure.

$$\frac{\mathrm{d}z}{\mathrm{d}x} = \sqrt{\frac{\omega^2 - f^2}{N^2 - \omega^2}} \qquad (R1)$$

(Changes are made in Lines 366, 372, and 375-376 in the marked-up manuscript version.)

13. Line 380 (Figure 16c-d): The authors gave two sub-figures to compare topographic steepness parameters for days 137 and 151. Unsurprisingly, they are very similar. In Lines 390-391, the authors talked about the similarity but do not explain why they are so similar. In fact, according to Eq. (3), the difference between the two is only in the stratification, which will not change significantly in just half a month, so the similarity is inevitable. This should have been explained in the text.

**Responses:** We thank the reviewer for this nice interpretation, we added this in the revised text.

14. The article uses a lot of formulas, and some of the math symbols are so similar that it's easy to get confused. For example, "c_n" and "c_p" are the eigen speed and phase speed (Line 412), "C_n" is topographic conversion term usually appears with energy exchange term "P_n" (Line 466), "p_n" is the pressure perturbation (Line 510). Then what's the meaning of "c_n" (Line517, A4) and "c_m" (Line 512, A3)?

**Responses:** We apologize for this information gap, $c_n$ and $c_m$ share the same meaning, $c_n$ stands for the eigen speed of the $n$th mode and $c_m$ stands for the eigen speed of the $m$th mode. The reason for using the "new" expression $c_m$ is to distinguish it from $c_n$, since the intermodal interaction involves the eigen speed of both the $n$th and $m$th modes. We checked all the equations in the paper to make sure the meaning of each variable is clear.

15. Line 517: should the "ω" in Eq. A4 be "ω_n"? The expression of this equation may be related to the modes, I think.

**Responses:** We understand the reviewer's concern. Vertical modal decomposition is usually used for internal waves with a fixed frequency (e.g., for SIT in this paper), and as a result, all decomposed modes also have a fixed frequency $\omega$.